# SiLVR🎲: A Simple Language-based Video Reasoning Framework

**Ce Zhang**                                                         *cezhang@cs.unc.edu*
*Department of Computer Science*
*UNC Chapel Hill*

**Yan-Bo Lin**                                                        *yblin@cs.unc.edu*
*Department of Computer Science*
*UNC Chapel Hill*

**Ziyang Wang**                                                     *ziyangw@cs.unc.edu*
*Department of Computer Science*
*UNC Chapel Hill*

**Mohit Bansal**                                                    *mbansal@cs.unc.edu*
*Department of Computer Science*
*UNC Chapel Hill*

**Gedas Bertasius**                                                   *gedas@cs.unc.edu*
*Department of Computer Science*
*UNC Chapel Hill*

**Reviewed on OpenReview:** `https://openreview.net/forum?id=mQZbh9Zlbw`

## Abstract

Recent advances in test-time optimization have led to remarkable reasoning capabilities in Large Language Models (LLMs), enabling them to solve highly complex problems in math and coding. However, the reasoning capabilities of multimodal LLMs (MLLMs) still significantly lag, especially for complex video-language tasks. To address this issue, we present SiLVR, a **Si**mple **L**anguage-based **V**ideo **R**easoning framework that decomposes complex video understanding into two stages. In the first stage, SiLVR transforms raw video into language-based representations using multisensory inputs, such as short clip captions and audio/speech subtitles. In the second stage, language descriptions are fed into a powerful reasoning LLM to solve complex video-language understanding tasks. To handle long-context multisensory inputs, we use an Adaptive Context Reduction scheme, which dynamically determines the temporal granularity with which to sample the tokens. Our simple, modular, and training-free video reasoning framework achieves the best-reported results on VideoMME (long), Video-MMLU, CGBench, and EgoLife. Furthermore, our empirical study focused on video reasoning capabilities shows that despite not being explicitly trained on video, strong reasoning LLMs can effectively aggregate multisensory input information from video, speech, and audio for complex temporal, causal, long-context, and knowledge acquisition reasoning tasks in video. More details can be found at `https://sites.google.com/cs.unc.edu/silvr`.

## 1 Introduction

Recent years have witnessed remarkable progress in general video understanding, with large multimodal models achieving strong performance on tasks such as video question-answering (VideoQA) (Team et al., 2023; Hurst et al., 2024; Bai et al., 2025; Zhang et al., 2024c), text-video retrieval (Zhao et al., 2023), and

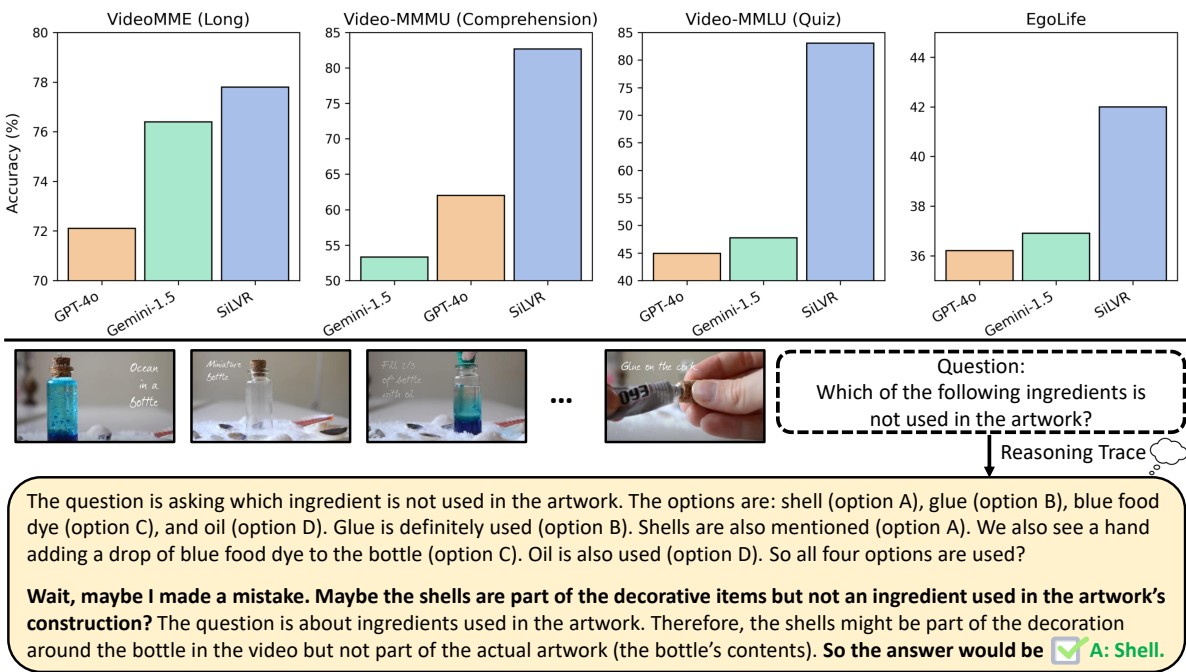

Figure 1: **Strong reasoning capabilities of SiLVR on complex video QA tasks.** SiLVR leverages recent advances in reasoning LLMs for complex video QA tasks. SiLVR achieves better performance than strong proprietary non-reasoning models (i.e., GPT-4o and Gemini-1.5) on benchmarks like VideoMME (long), Video-MMMU (Comprehension), Video-MMLU, and EgoLife, which include temporal, causal, long-context, and external knowledge reasoning tasks. The example reasoning trace shows SiLVR's capability to perform self-correction, in which it successfully identifies that shells are decorative rather than functional.

temporal localization (Huang et al., 2024; Ren et al., 2024; Wang et al., 2024c). Despite the remarkable progress, most existing methods struggle with complex video-language understanding tasks that require strong reasoning capabilities (e.g., temporal, causal, long-context, external knowledge, etc.). Following the success of reasoning LLMs (Guo et al., 2025; Jaech et al., 2024), several recent multimodal large language models (MLLMs) proposed reasoning frameworks for multimodal image/video recognition tasks (Liu et al., 2025; Fei et al., 2024; Wang et al., 2024d; Wu et al., 2025; Feng et al., 2025; Chen et al., 2025; Li et al., 2025; Wang et al., 2025). However, these methods either rely on high-quality Chain-of-Thought (CoT) data, which is expensive and time-consuming to collect, or require task-specific reward designs, leading to poor generalization. Moreover, such RL-based multimodal reasoning approaches are difficult to optimize and often require a large amount of resources for training. Lastly, many recently proposed RL post-training techniques lead to similar or sometimes even worse performance than standard Supervised Fine-tuning (SFT) approaches (Wang & Peng, 2025; Feng et al., 2025).

Motivated by the impressive reasoning abilities of recent LLMs (Guo et al., 2025; Jaech et al., 2024), we propose SiLVR, a simple, modular, and training-free language-based framework for complex video-language reasoning tasks. SiLVR decomposes video understanding into two stages:

- In the first stage, we convert raw videos into rich language-based descriptions. Specifically, we densely sample short clips from the input videos and use a pre-trained visual captioner (e.g., NVILA (Liu et al., 2024b)) to extract captions for each clip. Additionally, we use automatic speech recognition (ASR) tools to convert speech into language descriptions.
- In the second stage, we feed the rich language descriptions into a strong reasoning LLM (e.g. DeepSeek-R1 (Guo et al., 2025)) to solve complex video-language understanding tasks.

To address the issue of processing a large number of tokens in potentially hour-long videos, we propose a simple Adaptive Context Reduction scheme, which dynamically determines the temporal granularity with

which to sample the speech and video tokens. Such a context reduction scheme enables us to significantly reduce the number of input tokens to fit within the context length of an LLM, while maintaining strong reasoning performance.

Compared to prior MLLM-based video reasoning frameworks, SiLVR is simple, modular, training-free, and highly-performant. SiLVR achieves state-of-the-art results on multiple VideoQA benchmarks, including Video-MME (long), Video-MMLU, CGBench, and EgoLife. Additionally, SiLVR demonstrates strong spatiotemporal grounding ability for video QA tasks that require localizing relevant video segments. On CGBench, a large-scale grounded VideoQA benchmark, SiLVR outperforms the previous best method by a substantial margin of **6.1%** in mIoU. Additionally, our empirical study on video reasoning capabilities of our framework suggests that despite not being trained on videos, strong reasoning LLMs can successfully aggregate information from video, speech/audio for complex temporal, causal, long-context, and external knowledge reasoning tasks on video inputs.

While SiLVR is not based on any new complex design choices, it is simple, modular, training-free, and highly performant and generalizes to multiple complex video-language understanding tasks. We believe the simple yet effective design of SiLVR will enable the research community to build on our work and use our simple framework as a baseline to develop even more powerful video-language reasoning models.

## 2  Related Work

**Language Reasoning Models.** Recent work has significantly advanced the reasoning capabilities of LLMs through various strategies, such as Chain-of-Thought (Wei et al., 2022; Kojima et al., 2022), Self-Consistency (Wang et al., 2023), and Monte Carlo Tree Search based methods (Wan et al., 2024; Trinh et al., 2024; Xin et al., 2024). Recently, works such as DeepSeek-R1 (Guo et al., 2025) demonstrated that applying large-scale RL with accuracy and format rewards can induce emerging reasoning capabilities in LLMs. These RL-based methods have shown strong improvements in tasks such as mathematics (Zheng et al., 2021; Azerbayev et al., 2023) and code generation (Austin et al., 2021; Hendrycks et al., 2021). Motivated by these successes, we propose to take advantage of the strong reasoning ability of LLMs for complex video-language reasoning problems.

**Multimodal Reasoning Models.** There have been many efforts to augment MLLMs with reasoning capabilities. One line of work focuses on decomposing the reasoning process into multiple sub-problems (Shao et al., 2024; Zhang et al., 2023c; Xu et al., 2024; Zhang et al., 2024b). Motivated by the success of DeepSeek-R1 (Guo et al., 2025), another line of work explores RL to elicit the reasoning ability of the MLLMs (Huang et al., 2025; Shen et al., 2025; Yang et al., 2025b; Zhang et al., 2025; Ouyang, 2025; Peng et al., 2025). In the video domain, VideoCoT (Wang et al., 2024d), Video-of-Thought (Fei et al., 2024), and VideoEspresso (Han et al., 2025) propose to prompt the MLLMs with multiple reasoning steps before answering the question. In addition, multiple concurrent works propose to use GRPO to enhance video reasoning (Wu et al., 2025; Feng et al., 2025; Chen et al., 2025; Li et al., 2025; Wang et al., 2025). However, many of these RL-based methods demand substantial training computation, and achieve only marginal improvements or perform even worse than the SFT methods (Wang & Peng, 2025; Feng et al., 2025). Unlike these methods, SiLVR is simple, training-free, yet highly performant across a wide range of VideoQA benchmarks.

**Complex Video-Language Understanding.** A variety of benchmarks for complex video-language understanding have been proposed, with a focus on comprehensive evaluation of videos with different durations and questions spanning diverse categories (Fu et al., 2024; Li et al., 2024b; Liu et al., 2024a; Rawal et al., 2024; Zhou et al., 2024). In parallel, several benchmarks have been introduced to assess the reasoning capabilities of large video-language models (Hu et al., 2025; Song et al., 2025; Zhao et al., 2025; He et al., 2024). On the modeling side, recent video MLLMs adapt image MLLMs by fine-tuning additional modules for temporal modeling (Lin et al., 2023; Li et al., 2023; 2024a; Zhang et al., 2023b). Several follow-up works explored spatiotemporal token compression (Islam et al., 2025; Liu et al., 2024b; Bai et al., 2025; Shen et al., 2024; Shu et al., 2025), building hierarchical memory (Song et al., 2023; Jin et al., 2024; Islam et al., 2024), or incorporating information from external models (Tang et al., 2025; Wang et al., 2024a) for complex and long video understanding. Another line of work explores training-free frameworks that first convert raw videos into dense visual captions, then perform reasoning with off-the-shelf LLMs (Zhang et al., 2023a; Wang

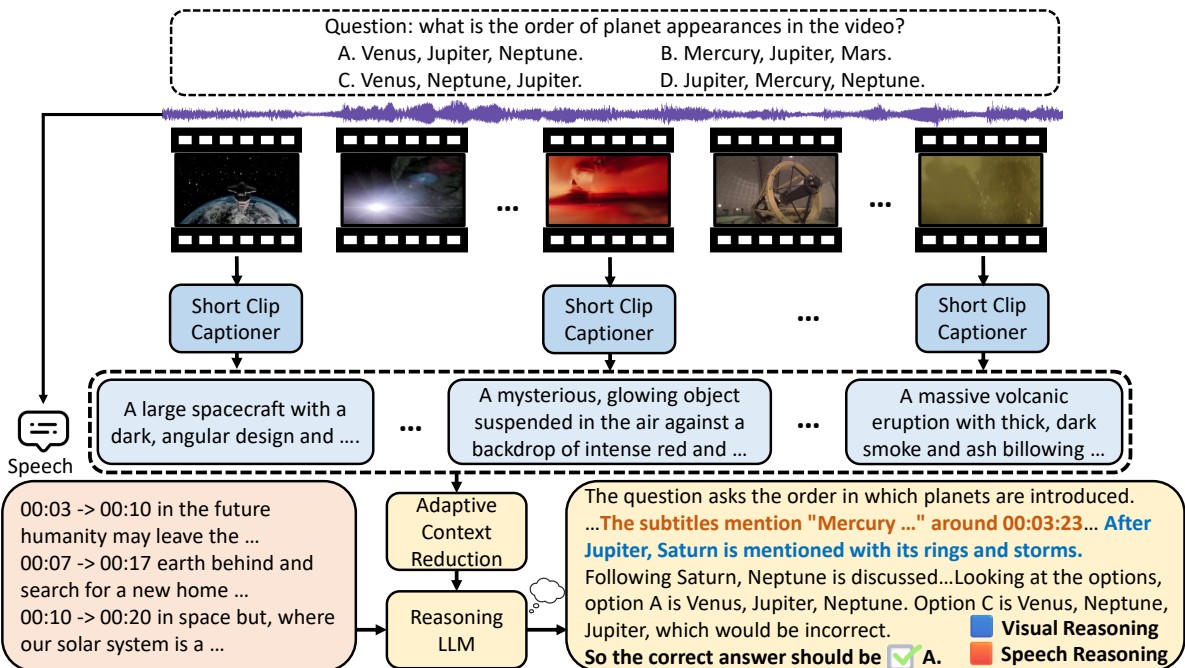

Figure 2: **Method overview.** SiLVR is a simple two-stage language-based video reasoning framework. **Top**: The video is segmented into short clips and paired with speech. A clip captioner processes each segment to generate visual descriptions. The speech is transcribed using ASR. **Bottom**: A reasoning LLM takes the question, transcribed speech, and dense visual descriptions compressed by Adaptive Context Reduction to perform complex video reasoning. In the shown example, SiLVR infers the correct order by integrating information across both visual and speech modalities. The model correctly identifies the sequence through reasoning and eliminating incorrect options.

et al., 2024b; Fan et al., 2024; Ma et al., 2024; Liao et al., 2024; Wang et al., 2024e; Min et al., 2024). In contrast to prior agent-based video understanding methods, SiLVR focuses on solving complex video-language reasoning problems with strong reasoning LLMs. Additionally, SiLVR employs a simple single-pass design, effectively integrates both visual and audio modality streams, and uses Adaptive Context Reduction to efficiently manage long input context.

## 3 Method

Our method decomposes video-language QA into two stages: 1) extracting visual captions and transcribing speech into text, and 2) performing language-based reasoning over the extracted textual descriptions. Such a decomposed video reasoning design offers several benefits: 1) **Simplicity**: SiLVR does not require complex RL-based video optimization or specialized modules for different tasks. 2) **Generalizability**: our method can be applied to a wide range of complex video-language tasks without task-specific fine-tuning. 3) **Modularity**: our method's modular design enables seamless use of powerful visual captioning models and strong reasoning LLMs. 4) **Flexibility**: SiLVR supports plug-and-play integration of different captioning models, speech recognition models, and LLMs. An overview of our method is illustrated in Figure 2.

### 3.1 Extracting Multimodal Descriptions

Given a video $V$, we first divide it into $N$ non-overlapping short clips $v = \{v_i\}_{i=1}^N$, where each clip $v_i \in \mathbb{R}^{T \times H \times W \times 3}$ contains $T$ frames of height $H$ and width $W$. Each clip is passed through a pretrained visual captioning model $\mathcal{M}$ to produce a caption $c_i$. The sequence of captions is denoted as $C = \{c_i\}_{i=1}^N$, forming a temporally ordered description of the visual content.

---

**Algorithm 1** Adaptive Context Reduction

---

**Require:** Video $V$, Question $Q$, LLM $F$, Captioner $M$, Speech Recognition Model $W$, Initial Clip Length $L$
  1: $S \leftarrow \text{extractSubtitles}(V, W)$
  2: $limit \leftarrow \text{getContextLength}(F)$
  3: **while** True **do**
  4:    $v \leftarrow \text{divideVideo}(V, L)$
  5:    $C \leftarrow \text{generateCaptions}(v, M)$
  6:    $Z \leftarrow \text{concat}(S, C)$
  7:    **if** $\text{countTokens}(Z) > limit$ **then**
  8:       $L \leftarrow L \times 2$
  9:    **else**
 10:       **break**
 11:    **end if**
 12: **end while**
 13: **return** $\text{answer}(Z, Q, F)$

---

In parallel, we apply an ASR model $\mathcal{W}$ to convert the speech into a sequence of textual descriptions $S = \{s_j\}_{j=1}^{K}$, where $s_j$ is a timestamped transcription of a spoken segment. $K$ denotes the total number of segments, which is determined by an ASR model. We then concatenate $S$ and $C$ one after the other to form a rich, language-based description of the video (including audio/speech) and feed them into a reasoning LLM as described next.

### 3.2 Language-Based Reasoning with Adaptive Context Reduction

To answer a question $Q$ about the video, we feed the video captions $C$ and speech transcriptions $S$ along with $Q$ into a reasoning LLM. We design several prompts to guide the LLM to reason jointly over visual and speech information (for complete prompts see Supplementary Material). Unlike prior video reasoning approaches, SiLVR performs reasoning entirely in the language space. However, the limited context window of LLMs poses a significant challenge when processing long videos with rich multimodal content. To address this issue, we introduce a simple adaptive context reduction scheme (see Algorithm 1). Our Adaptive Context Reduction scheme dynamically adjusts the temporal granularity for sampling video clips. Specifically, it starts with a small clip length and progressively increases it to reduce the total number of generated tokens. This allows us to effectively fit the input tokens within the LLM's context window for videos of varying durations while maintaining strong video reasoning performance.

### 3.3 Implementation Details

We use NVILA (Liu et al., 2024b) as the default visual captioning model. We use our Adaptive Context Reduction scheme for all videos as described above. For speech transcription, we use Whisper-large-v3 (Radford et al., 2022). Due to its strong reasoning performance, we use DeepSeek-R1 (Guo et al., 2025) as the default LLM and set the temperature to 1.0 for all experiments. We use the official evaluation code provided by each benchmark, or use LMMs-Eval (Zhang et al., 2024a) when the official code is unavailable. Additional implementation details are provided in the supplementary materials.

## 4 Experiments

### 4.1 Benchmarks and Evaluation Metrics

We conduct experiments on eight complex video-language understanding benchmarks: Video-MMMU (Hu et al., 2025), Video-MMLU (Song et al., 2025), MMVU (Zhao et al., 2025), MMWorld (He et al., 2024), VideoMME (Fu et al., 2024), CGBench (Chen et al., 2024), EgoLife (Yang et al., 2025a) and CinePile (Rawal et al., 2024). Following Video-R1 (Feng et al., 2025), we group these benchmarks into two categories: Reasoning Benchmarks and General Benchmarks. The reasoning benchmarks include Video-MMMU, Video-

Table 1: **Main results.** We evaluate our method on a set of video reasoning benchmarks (Video-MMMU, Video-MMLU, MMVU, MMWorld) and general video benchmarks (VideoMME, CGBench, EgoLife, CinePile). We use the comprehension split of Video-MMMU and the long split of VideoMME (with subtitles). Based on these results, we observe that SiLVR achieves the best-reported results on Video-MMLU, VideoMME (long split, with subtitles), CGBench, and EgoLife, outperforming strong proprietary models such as Gemini 2.0 and GPT-4o. We **bold** and underline the best and the second best models in each benchmark, respectively.

| Model | Video Reasoning Benchmarks | | | | General Video Benchmarks | | | |
|---|---|---|---|---|---|---|---|---|
| | Video-MMMU | Video-MMLU | MMVU | MMWorld | VideoMME | CGBench | EgoLife | CinePile |
| *Proprietary Models* | | | | | | | | |
| Gemini 1.5 Flash | 49.0 | 47.8 | 58.8 | - | 68.8 | 33.5 | - | 58.8 |
| Gemini 1.5 Pro | 53.5 | - | 65.4 | 51.0 | 77.4 | 37.8 | 36.9 | **60.1** |
| Gemini 2.0 Flash | - | - | 65.9 | - | - | - | - | - |
| Gemini 2.5 Flash | 79.2 | - | - | - | - | - | - | - |
| Gemini 2.5 Pro | **83.6** | - | - | - | - | - | - | - |
| GPT-4o | 62.0 | 44.9 | 67.4 | **62.5** | 72.1 | 44.9 | 36.2 | 56.1 |
| Claude 3.5 Sonnet | 75.7 | 71.3 | 65.2 | 54.5 | - | 40.3 | - | - |
| Kimi-k1.6 | 76.7 | - | - | - | - | - | - | - |
| OpenAI o1 | - | - | **75.5** | - | - | - | - | - |
| *Open-Source LMMs* | | | | | | | | |
| LLaVA-OV-7B | 31.0 | 33.4 | 37.9 | - | - | 30.9 | 30.8 | 49.3 |
| VideoLLaMA3-7B | 46.0 | - | 45.0 | - | 61.0 | - | - | - |
| Aria | 53.0 | - | 49.3 | - | 66.3 | - | - | - |
| DeepSeek-VL2 | - | - | 52.1 | - | - | - | - | - |
| Qwen-2-VL-72B | - | - | 50.3 | - | 74.3 | 45.3 | - | - |
| Qwen-2.5-VL-7B | 50.4 | 32.9 | - | - | - | - | - | - |
| Qwen-2.5-VL-7B | 61.0 | 40.5 | - | - | - | - | - | - |
| Qwen-3-VL-8B | 72.8 | - | - | - | - | - | - | - |
| Qwen-3-VL-32B | 79.0 | - | - | - | - | - | - | - |
| Video-R1 | 52.4 | - | 52.1 | - | - | - | - | - |
| SiLVR (ours) | 82.7 | **83.1** | 68.2 | 59.9 | **77.7** | **51.8** | **42.0** | 59.4 |

MMLU, MMVU, and MMWorld, which primarily evaluate the reasoning capabilities of large video-language models. Specifically, Video-MMMU and Video-MMLU focus on lecture-based video understanding, where the model must extract knowledge from lecture videos to answer the questions. MMVU requires models to apply domain-specific knowledge and perform expert-level reasoning to analyze specialized-domain videos. MMWorld focuses on a diverse set of reasoning questions (e.g., counterfactual thinking, future prediction, etc.) across videos from seven broad disciplines. The other four benchmarks (i.e., VideoMME, CGBench, EgoLife, and CinePile) are general video-language benchmarks, which contain various types of questions and offer a comprehensive assessment of the video-language models. Specifically, VideoMME includes three splits (short, medium, and long) based on the duration of the video. CGBench and EgoLife are designed for long video understanding, with an average video duration of more than an hour. We focus on VideoQA for all benchmarks and report QA accuracy as our primary evaluation metric. Additionally, we conduct Grounded VideoQA experiments on CGBench to assess the model's temporal grounding ability and use the mean Intersection over Union (mIoU) to evaluate the results.

## 4.2 Main Results

**Video Reasoning Benchmarks.** We present our results on video reasoning benchmarks on the left side of Table 1. Our results indicate that SiLVR achieves the strongest performance across all video reasoning benchmarks among open-source models such as Qwen-3-VL-72B and Video-R1. Additionally, SiLVR achieves the best performance on Video-MMLU, outperforming the previous state-of-the-art model Claude 3.5 Sonnet by a substantial **11.8%**. On Video-MMMU, SiLVR outperforms strong proprietary models such as Gemini

Table 2: **Performance comparison between one reasoning (DeepSeek-R1) and two non-reasoning (Llama 4, DeepSeek-V3) LLMs.** Using DeepSeek-R1 reasoning LLM leads to significantly better results over non-reasoning LLMs (Llama 4 and DeepSeek-V3). However, we also observe that the average gain on video reasoning benchmarks (VideoMMMU, VideoMMLU, MMVU, MMWorld) is significantly larger than on general video benchmarks (VideoMME, CGBench, EgoLife, CinePile). These results demonstrate that the strong reasoning ability of DeepSeek-R1 is crucial for solving complex video reasoning tasks.

| Model | Video Reasoning Benchmarks | | | | General Video Benchmarks | | | |
|---|---|---|---|---|---|---|---|---|
| | VideoMMMU | VideoMMLU | MMVU | MMWorld | VideoMME | CGBench | EgoLife | CinePile |
| *Non-Reasoning LLMs* | | | | | | | | |
| SiLVR (Llama 4) | 56.3 | 57.2 | 60.6 | 57.2 | 67.8 | 53.2 | 38.5 | 45.6 |
| SiLVR (DeepSeek-V3) | 65.7 | 74.7 | 62.9 | 58.7 | 75.0 | 50.2 | **42.2** | 56.7 |
| *Reasoning LLMs* | | | | | | | | |
| SiLVR (DeepSeek-R1) | **82.7** | **83.1** | **68.2** | **59.9** | **77.7** | **59.4** | 42.0 | **51.8** |
| | Average Gain over DeepSeek-V3: **+8.0** | | | | Average Gain over DeepSeek-V3: **+3.8** | | | |

2.5 Flash by **3.5%**. Furthermore, on MMVU, we observe that our modular framework, with DeepSeek-R1 as the LLM, outperforms the unified multimodal model DeepSeek-VL2 by a significant margin of **15.9%**. These results suggest that despite the simplicity of our approach, it delivers strong performance across a wide range of video-language reasoning tasks.

**General Video Benchmarks.** We present our results on general video benchmarks on the right side of Table 1. Based on these results, we observe that SiLVR achieves state-of-the-art performance on three general benchmarks: VideoMME (long split, with subtitles), CGBench, and EgoLife. Specifically, on VideoMME and EgoLife, SiLVR outperforms the prior best performing method Gemini 1.5 Pro by **0.3%** and **5.1%**, respectively. On CGBench, SiLVR achieves 51.8% accuracy, outperforming the previous state-of-the-art method Qwen-2-VL-72B by a significant **6.9%** margin. SiLVR also surpasses strong proprietary models, outperforming GPT-4o by **6.9%** and Claude 3.5 Sonnet by **11.5%** on CGBench. Additionally, it is worth noting that VideoMME (long), EgoLife, and CGBench are designed for very long-form video understanding, with average video durations exceeding 60 minutes. Our strong results demonstrate that SiLVR is highly effective in comprehending long videos.

### 4.3 Reasoning Analysis

In this section, we conduct a more in-depth analysis of the video reasoning capabilities of our approach. To do this, we systematically compare the performance of our framework when using reasoning (e.g., DeepSeek-R1) vs. non-reasoning (e.g., Llama 4 and DeepSeek-V3) LLMs across multiple benchmarks. Additionally, we break down the performance of our approach across different types of video reasoning questions (e.g., temporal, causal, long-context, knowledge acquisition, etc.).

**Reasoning vs Non-Reasoning LLMs.** To study the impact of a strong reasoning LLM within our framework, we compare the performance of our method when using one reasoning LLM (DeepSeek-R1) vs. two non-reasoning LLMs (Llama 4 and DeepSeek-V3). The results are presented in Table 2. Our results suggest several interesting trends. First, we observe that DeepSeek-R1 consistently outperforms Llama 4 across all benchmarks, indicating that it is a much stronger LLM than Llama 4. Second, we note that using DeepSeek-R1 leads to much larger performance gains on the reasoning benchmarks, where DeepSeek-R1 surpasses DeepSeek-V3 by a substantial **17.0%** on Video-MMMU and **8.4%** on Video-MMLU with an average improvement of **8.0%** on all video reasoning benchmarks. In contrast, while DeepSeek-R1 also produces better results on general video benchmarks, the improvements over DeepSeek-V3 are much smaller (i.e., average improvement of **3.8%** on general video benchmarks vs. **8.0%** on the reasoning benchmarks). These results suggest that the strong reasoning ability of DeepSeek-R1 is critical for solving complex video reasoning

Table 3: **Comparison with agent-based methods on VideoMME (long).** Our method achieves consistently higher performance than all other agent-based baselines, indicating its effectiveness for long-form video reasoning.

| Method | Visual Captioner | LLM | Accuracy |
|--------|------------------|-----|----------|
| VideoAgent | CogAgent | GPT-4 | 46.4 |
| DrVideo | LLaVA-NeXT | GPT-4 | 51.7 |
| VideoTree | LLaVA1.6-7B | GPT-4o | 54.2 |
| VCA | GPT-4o | GPT-4o | 56.3 |
| SiLVR (ours) | NVILA-7B | DeepSeek-R1 | **62.7** |

tasks and that our framework's simple and modular design allows us to take full advantage of DeepSeek-R1's strong reasoning abilities on these complex video reasoning problems.

**Performance Breakdown Across Different Tasks.** In Figure 3, we report the performance gains of using a reasoning LLM (DeepSeek-R1) over a non-reasoning LLM (Llama 4) for different question categories on VideoMME, which contains 12 manually annotated categories. The four categories that we report on the left of Figure 3, belong to reasoning (e.g., temporal, spatial, object, and action reasoning). The other eight categories are classified as non-reasoning and require general perception capabilities (e.g., action recognition, OCR, etc.). Based on the results in Figure 3, we observe that compared to Llama 4, using DeepSeek-R1 achieves a significantly larger improvement on reasoning questions (a gain of $+11.1\%$) compared to non-reasoning questions (a gain of $+4.9\%$). This result is consistent with our observations in Table 2, which confirms that reasoning LLMs bring greater benefits for tasks that require complex reasoning.

## 4.4 Comparison with Agent-based Methods

In Table 3, we compare our method with other agent-based approaches on the long split of VideoMME (without subtitles). The baseline methods include VideoAgent Wang et al. (2024b), VideoTree Wang et al. (2024e), DrVideo Ma et al. (2024), and VCA Yang et al. (2025c). VideoAgent Wang et al. (2024b) iteratively selects frames and gathers visual information before answering the query. VideoTree Wang et al. (2024e) builds a hierarchical and query-adaptive tree of keyframes to efficiently summarize long videos for LLM reasoning. DrVideo Ma et al. (2024) formulates long-video QA as iterative video-document construction, where the model retrieves key frames and progressively generates the a textual description for reasoning. VCA Yang et al. (2025c) employs a reward-driven search strategy to explore informative video segments. All baseline methods require multiple rounds of interaction between a heavy-weight LLM while our method requires only a single LLM reasoning call. From Table 3, we can see that SiLVR substantially outperforms all other agent-based methods, confirming its superior design for long-form video reasoning.

## 4.5 Efficiency Analysis

In Table 4, we analyze the inference efficiency of our framework and compare it against prior video understanding models, including two video-native models (Qwen-2.5-VL, Video-R1) and two agent-based methods (DrVideo (Ma et al., 2024), VideoTree (Wang et al., 2024e)). We follow the evaluation setup of VideoTree (Wang et al., 2024e), which excludes ASR inputs, and measure the average processing time per video on the VideoMME (Long) benchmark. To ensure fairness, we run all methods on a single A6000, except for the Qwen-2.5-VL-7B (768-frame) setting, which requires four A6000 GPUs due to its high memory demand. In addition to the best-performing variant of our method (SiLVR-best), we also report a more efficient variant (SiLVR-fast), which uses a larger clip length to reduce the number of sampled clips and generated captions.

Table 4: **Efficiency comparison with other methods on VideoMME (long).** To ensure fair comparison, we follow the evaluation setup of VideoTree and use the same GPUs for all methods. Both SiLVR-best and SiLVR-fast significantly outperforms Qwen-2.5-VL, Video-R1, VideoTree and DrVideo. Notably, SiLVR-fast runs **1.4x** faster than Qwen-2.5-VL (768 frames), **5.3x** faster than DrVideo, and **2.0x** faster than VideoTree. These results demonstrate that SiLVR is both efficient and effective in complex video reasoning tasks.

| Method | Runtime (s) ↓ | Accuracy ↑ |
|---|---|---|
| *Video-native Models* | | |
| Qwen-2.5-VL 7B (32 frames) | **5.2** | 44.7 |
| Qwen-2.5-VL 7B (768 frames) | 115.8 | 50.8 |
| Video-R1 (32 frames) | 14.2 | 50.2 |
| *Agent-based Methods* | | |
| DrVideo | 446 | 51.7 |
| VideoTree | 162 | 54.2 |
| SiLVR-fast (ours) | 83 | 57.2 |
| SiLVR-best (ours) | 442 | **62.7** |

Table 5: **Results on knowledge acquisition and temporally grounded QA tasks.** SiLVR achieves the highest $\Delta_{\text{knowledge}}$ on VideoMMMU and the best mIoU on CGBench.

| Model | VideoMMMU ($\Delta_{\text{knowledge}}$) | CGBench (mIoU) |
|---|---|---|
| Qwen-2.5-VL-72B | 9.7 | - |
| Gemini-1.5 Pro | 8.7 | 3.85 |
| Claude-3.5 Sonnet | 11.4 | 4.17 |
| GPT-4o | 15.6 | 5.73 |
| VideoMind-7B | - | 7.10 |
| SiLVR (ours) | **17.2** | **11.84** |

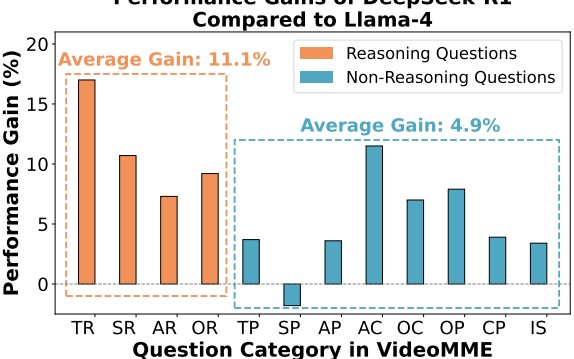

Figure 3: **Performance breakdown across different question categories.** Using DeepSeek-R1 as an LLM yields larger gains on reasoning questions (**+11.1%**) than general perception questions (**+4.9%**). Full category names are in the supplementary materials (Table 18).

From Table 4, we can observe that both SiLVR-best and SiLVR-fast achieve higher accuracy than all other methods. Specifically, SiLVR-best outperforms video-native models Qwen-2.5-VL 7B (768 frames) and Video-R1 by **11.9%** and **12.5%**, respectively. SiLVR-best also outperforms agent-based methods DrVideo (Ma et al., 2024) and VideoTree (Wang et al., 2024e) by a significant margin of **11.0%** and **8.4%**, respectively. Additionally, SiLVR-fast is highly efficient, running **1.4x** faster than Qwen-2.5-VL-7B (768 frames), **5.3x** faster than DrVideo and **2.0x** faster than VideoTree, while still achieving **+6.4%**, **+5.5%** and **+3.0%** higher accuracy, respectively. Although Qwen-2.5-VL 7B (32 frames) adopts a sparse frame sampling strategy to achieve high efficiency, this design leads to degraded performance (**-12.5%**) compared with SiLVR-fast. These results demonstrate that our framework is able to achieve an effective balance between efficiency and accuracy, making it both effective and efficient in complex video reasoning tasks.

### 4.6 Results on Other Tasks

**Knowledge Acquisition from Videos.** We also evaluate our method on the novel knowledge acquisition task on Video-MMMU (Hu et al., 2025). The task requires models to answer questions both before and after watching a reference lecture video, with the goal of measuring how much knowledge the model gains from the

Table 6: **Impact of ASR.** We remove the ASR module and evaluate our method on eight benchmarks. Incorporating ASR consistently improves performance, highlighting the importance of speech information in video reasoning.

| Method | Video-MMMU | Video-MMLU | MMVU | MMWorld | VideoMME | CGBench | EgoLife | CinePile |
|---|---|---|---|---|---|---|---|---|
| SiLVR (w/ ASR) | **82.7** | **83.1** | **68.2** | **59.9** | **77.7** | **51.8** | **42.0** | **59.4** |
| SiLVR (w/o ASR) | 70.7 | 78.2 | 61.7 | 57.4 | 62.7 | 40.9 | 35.1 | 47.4 |

Table 7: **Token reduction analysis.** Accuracy on VideoMME (overall) when selectively dropping speech vs. visual caption tokens. Based on these results, we observe that speech tokens are more informative than visual caption tokens.

| Dropping Rate | | Average Context Length | Accuracy |
|---|---|---|---|
| Subtitles | Captions | | |
| 50% | - | 4.3k | 65.3 |
| 75% | - | 2.6k | 56.0 |
| - | 50% | 7.2k | 68.9 |
| - | 75% | 6.0k | 67.7 |
| No Compression | | 9.3k | 70.3 |

video. The metric for the knowledge acquisition task is defined as:

$$\Delta_{\text{knowledge}} = \frac{\text{Acc}_{\text{post}} - \text{Acc}_{\text{pre}}}{100\% - \text{Acc}_{\text{pre}}} \times 100\% \tag{1}$$

where $\text{Acc}_{\text{pre}}$ and $\text{Acc}_{\text{post}}$ denote the accuracy before and after watching the video, respectively.

Our results in Table 5 show that SiLVR achieves **17.2%** in $\Delta_{\text{knowledge}}$, outperforming the prior best method GPT-4o by **1.6%**. SiLVR also outperforms strong proprietary models such as Gemini-1.5 Pro and Claude-3.5 Sonnet by **8.5%** and **5.8%**, respectively. These results demonstrate that SiLVR is not only effective in complex video reasoning, but also has strong knowledge acquisition capabilities.

**Temporally Grounded QA.** In Table 5, we also present our results on the temporally grounded QA task on CGBench (Chen et al., 2024). The task requires the model to temporally localize relevant video segments needed to answer the question (usually less than 10 seconds) in long videos that span over 60 minutes. Our results in Table 5 show that SiLVR achieves the highest performance in mIoU, outperforming concurrent work VideoMind (Liu et al., 2025) by a notable **4.74%**. In addition, SiLVR also outperforms GPT-4o and Claude-3.5 Sonnet by **6.11%** and **7.67%**, respectively. These results suggest that SiLVR can correctly answer complex questions and temporally ground the answer to relevant segments in the video, which improves interpretability in video reasoning.

### 4.7 Ablation Studies

Unless otherwise specified, all ablation experiments are conducted on the VideoMME benchmark.

**Impact of the ASR.** To analyze the impact of ASR, we remove the ASR module of our method and evaluate our method on eight benchmarks. The results are shown in Table 6. From the table, we observe that ASR plays a crucial role across many benchmarks. On the lecture understanding benchmark Video-MMMU, incorporating ASR leads to a significant 12.0% accuracy gain, which aligns with the intuition that spoken content in lectures provides essential cues. Additionally, adding ASR leads to a 12.0% accuracy improvement on the movie understanding benchmark CinePile. Furthermore, in EgoLife and CGBench, which focus on analyzing short clues within long videos, ASR also provides notable gains. Finally, on general-purpose

Table 8: **Performance of Adaptive Context Reduction (ACR) vs. fixed clip length baselines.** ACR achieves the highest accuracy on VideoMME (overall), outperforming the best fixed-length baseline (8s) by **2.5%**. These results suggest that ACR effectively reduces redundant tokens while preserving strong performance.

| Clip Length (s) | ACR | 1 | 2 | 4 | 8 | 64 |
|---|---|---|---|---|---|---|
| Accuracy | **76.7** | 59.9 | 61.3 | 68.5 | 74.2 | 70.3 |

Table 9: **Comparison between two context-reduction strategies on VideoMME (overall).** Reducing caption word limits and reducing the number of segments (Adaptive Context Reduction) show similar performance.

| Variant | Accuracy |
|---|---|
| Change Word Limit | 76.6 |
| Change Segment Number (ACR) | **76.7** |

benchmarks like VideoMME and MMVU, adding ASR improves performance by a large margin, demonstrating that speech complements visual signals in multiple video understanding tasks.

**Speech vs. Visual Caption Token Importance.** To evaluate the relative contribution of visual and audio information, we vary the fraction of tokens from speech transcripts and video captions and report the QA performance on VideoMME. As shown in Table 7, the reduction of 50-75% speech tokens (while keeping all visual caption tokens) leads to a significant decrease in performance (**11.4%-20.7%**). In comparison, dropping the same fraction of visual caption tokens (while keeping all speech tokens) results in a much smaller performance drop (**7.8%-9.0%**). There results indicate that speech tokens are more informative than visual caption tokens.

**Analysis of Adaptive Context Reduction.** In Table 8, we compare Adaptive Context Reduction (ACR) with several static baselines that use fixed video clip lengths. Among all baselines, the variant that uses an 8-second clip length achieves the highest accuracy of 74.2%. We note that a shorter clip variant (e.g., 1s) generates a large number of captions for long videos, which often exceeds the context window of the LLMs, thus leading to degraded performance. In contrast, a longer clip variant (e.g., 64s) reduces the number of captions at the cost of sacrificing the granularity of visual information, which also leads to lower accuracy. Compared to these static baselines, our proposed ACR consistently outperforms all fixed clip length baselines, surpassing the best-performing variant (8s) by a significant margin of **2.5%**. These results demonstrate that ACR effectively reduces redundant tokens by adaptively adjusting the clip length, offering flexibility and strong performance.

**Context Reduction Method Ablation.** In addition to Adaptive Context Reduction, we introduce an alternative strategy for reducing the input context length of the LLM. Specifically, we modify the captioning prompt to constrain each caption to 40, 20, or 10 words. When the context limit is exceeded, we progressively reduce the per-segment caption length from 40 words to 20 words, and then to 10 words. The results are shown in Table 9. From the results we can see that changing word limit yields similar results to changing the segment number.

**Comparison between Uniform and Non-uniform Sampling.** We experiment with the non-uniform frame sampling strategy used in VideoTree Wang et al. (2024e) and compare it against uniform sampling in our framework. Specifically, VideoTree extracts CLIP features for each frame, performs k-means clustering into 32 groups, and selects the cluster centers as representative frames. We use these non-uniformly sampled frames in our framework and compared them to uniform sampling on VideoMME (Long, without subtitles). The results are shown in Table 10. From the table, we can observe that non-uniform sampling improves our

Table 10: **Comparison between uniform and non-uniform sampling.** Following VideoTree, we evaluate our method on VideoMME (Long). We apply the non-uniform sampling strategy of VideoTree to our framework. Non-uniform sampling improves our method by 0.8%, showing that our method can benefit from advanced sampling techniques.

| Sampling Method | Accuracy |
|---|---|
| Uniform Sampling | 76.8 |
| Non-Uniform Sampling | **77.6** |

Table 11: **Effects of LLM input context length on VideoMME (overall).** Increasing the LLM input context consistently improves the performance, showing that the benefit of more information outweighs the increased context length in our framework.

| Average Context Length | Accuracy |
|---|---|
| 9.3k | 70.3 |
| 18.2k | 74.2 |
| 29.5k | 76.7 |

Table 12: **Performance with different visual captioners on VideoMME.** Qwen-2.5-VL 72B achieves the best overall accuracy. We use NVILA 7B for all experiments because it provides the best accuracy-cost trade-off.

| Captioner | Size | Overall | Short | Medium | Long |
|---|---|---|---|---|---|
| LLaVA-OV | 7B | 67.2 | 57.7 | 68.1 | 75.9 |
| NVILA | 7B | 70.3 | 63.2 | 70.4 | 77.3 |
| Qwen-2.5-VL | 7B | 70.9 | 63.8 | 72.9 | 76.1 |
| Qwen-2.5-VL | 72B | 71.2 | 65.0 | 72.2 | 76.4 |
| Qwen-3-VL | 3B | 71.0 | 63.5 | 70.9 | 78.6 |
| NVILA + Qwen-2.5-VL | 7B + 7B | **74.1** | **68.3** | **74.6** | **79.4** |

Table 13: **Performance of our framework with different LLMs on VideoMME.** Llama-4 Maverick achieves 66.2% overall accuracy, providing an effective trade-off between model sizes and performance. DeepSeek R1 achieves the highest overall accuracy, outperforming DeepSeek V3 and GPT-4.1 by **3.5%** and **0.8%**, respectively.

| LLM | Size | Overall | Short | Medium | Long |
|---|---|---|---|---|---|
| Llama-4-Scout | 17B | 63.0 | 56.7 | 64.4 | 67.8 |
| Llama-4-Maverick | 17B | 66.2 | 57.2 | 68.3 | 73.0 |
| DeepSeek V3 | 685B | 66.8 | 56.0 | 69.1 | 75.3 |
| DeepSeek R1 | 685B | 70.3 | 63.2 | 70.4 | 77.3 |
| GPT-4o | - | 67.3 | 57.0 | 69.2 | 75.8 |
| GPT-4.1 | - | 69.5 | 59.6 | 71.1 | 77.9 |
| GPT-5 | - | **73.9** | **64.7** | 75.1 | 81.9 |
| Gemini-2.5-Pro | - | 73.1 | 60.3 | **75.3** | **83.8** |

method by **0.8%**, demonstrating that our method can immediately benefit from more advanced, non-uniform sampling techniques.

**Effects of LLM Context Length.** Prior work suggests that excessively long inputs to the LLM can harm reasoning quality Levy et al. (2024); Kahatapitiya et al. (2025). In our setting, increasing the number of sampled clips provides more fine-grained visual information but also proportionally increases the context length, potentially affecting LLM performance. To study this trade-off, we vary the clip length to control the number of sampled clips and thus, the corresponding number of generated captions, which directly determines the total input context length to the LLM. For each configuration, we measure the average context length and evaluate the overall accuracy on VideoMME. The results are shown in Table 11. From the table, we

observe that increasing the context length consistently improves performance in our setting, suggesting that the benefits of providing more fine-grained visual information outweigh the potential drawbacks of longer input sequences.

**Visual Captioning Model.** Next, we study the effect of different visual captioners. As shown in Table 12, NVILA 7B and Qwen-2.5-VL 7B achieve similar performance, outperforming LLaVA-OV 7B by **2.9%** and **3.7%**, respectively. We also observe that Qwen-2.5-VL 72B achieves the higher overall accuracy compared with Qwen-2.5-VL 7B, likely due to the larger LLM (72B), which leads to higher-quality captions. Since NVILA 7B is faster than Qwen-2.5-VL 7B and achieves similar performance, we use NVILA 7B for all experiments. We do not use Qwen-2.5-VL 72B due to the prohibitive computational cost. We also observe that SiLVR remains effective and achieves a strong result (71.0%) even with a 3B parameter captioner (Qwen-3-VL 3B). Additionally, we explore the ensemble of two captioners (NVILA-7B + Qwen-2.5-VL-7B) by concatenating the generated captions from each model. As shown in the table, the ensemble captioner improves overall accuracy by **3.8%** and **3.2%** over NVILA-7B and Qwen-2.5-VL-7B individually, showing that our framework consistently benefits from stronger captioners.

**Different LLMs.** Lastly, in Table 13 we study the effect of different LLMs, including Llama-4-Scout 17B, Llama-4-Maverick 17B, DeepSeek V3, DeepSeek R1, GPT-4o, GPT-4.1, GPT-5, and Gemini-2.5-Pro. Our results indicate that Gemini-2.5-Pro, as the LLM backbone, achieves the highest accuracy on the long split of VideoMME, outperforming GPT-5 by a significant margin (**1.9%**). GPT-5 achieves the best overall accuracy on VideoMME, outperforming Gemini-2.5-Pro by **0.8%**. These results indicate that stronger LLMs consistently leads to better performance in our framework. Furthermore, we note that Llama-4-Maverick achieves 66.2% accuracy with only 17B parameters, offering an effective trade-off between model size and performance. Lastly, we observe that DeepSeek R1 outperforms DeepSeek V3 by a significant **3.5%**, highlighting the effectiveness of using a reasoning LLM within our framework.

### 4.8 Qualitative Results

We present two reasoning traces of SiLVR in Figure 4 and Figure 5. In Figure 4, the video showcases the Tesla Cybertruck, and the question asks about the size of its rear touchscreen display. Since the display appears only briefly, the question is particularly challenging. As shown in the figure, SiLVR first recognizes the vehicle as a Tesla Cybertruck. Leveraging both visual cues and the LLM's prior knowledge, SiLVR then correctly infers that the size of the rear touchscreen is 15 inches. This example demonstrates SiLVR's capability to incorporate visual information with the LLM's prior knowledge for complex video reasoning. Figure 5 shows a chemistry tutorial video and a follow-up question. SiLVR initially makes a tentative prediction but does not terminate the reasoning process immediately. Instead, it continues to validate the predicted answer through step-by-step reasoning. These two examples highlight that SiLVR is effective across diverse video domains, highlighting the generalizability of our proposed method. Additional qualitative analyses are provided in the supplementary materials.

## 5 Conclusion

We present SiLVR, a simple, modular, and training-free language-based video reasoning framework. SiLVR achieves state-of-the-art performance on VideoMME (long), Video-MMLU, CGBench, and Ego-Life. SiLVR also achieves strong results in video-based knowledge acquisition and temporally grounded QA tasks, demonstrating strong generalization. Lastly, we systematically analyze the reasoning capabilities of SiLVR and perform ablations on several key design choices. We encourage the research community to build on our simple yet effective video reasoning framework and hope that it will inspire new ideas in video reasoning research.

## Limitations

As with most modular frameworks, the performance of our method depends on its individual modules. On the visual perception side, our method relies on the visual captioning model, which may produce hallucinations

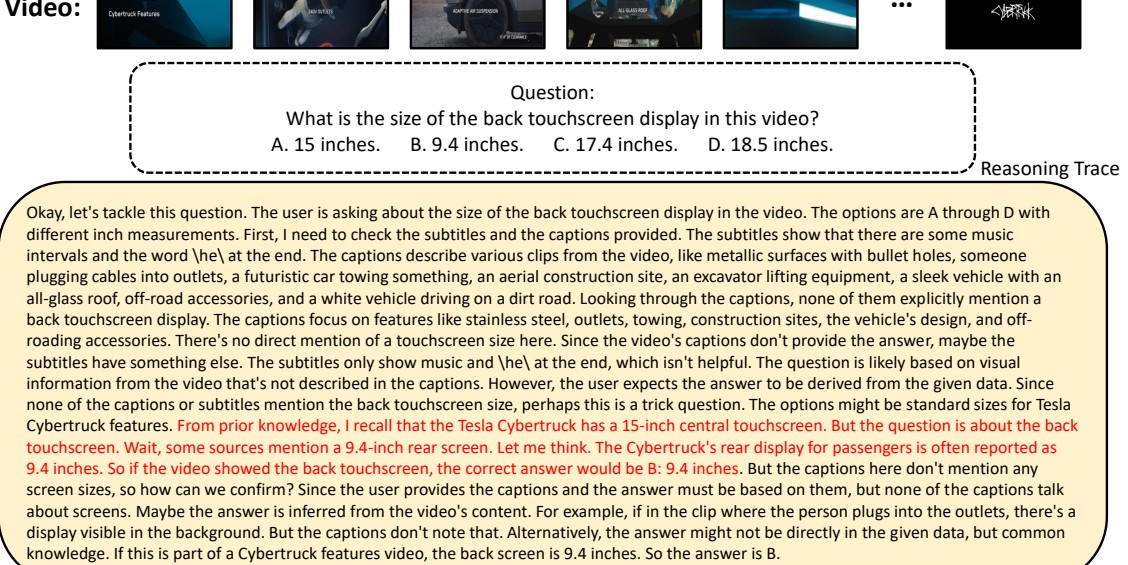

Figure 4: **Example 1 of SiLVR's reasoning trace.** The question asks about the size of the back touchscreen in the car shown in the video. The visual captioning module of SiLVR fails to capture the details about the touchscreen, which appears briefly for only about one second. However, by identifying the vehicle type and leveraging external knowledge from the LLM, SiLVR infers the correct answer.

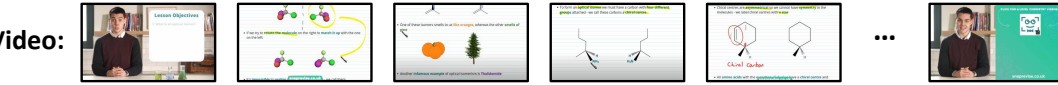

Figure 5: **Example 2 of SiLVR's reasoning trace.** Through step-by-step reasoning, SiLVR is capable of solving complex domain-specific questions. Notably, SiLVR does not immediately terminate the reasoning process upon reaching a plausible answer. Instead, it continues to verify the correctness of the generated answer before finalizing its response.

or descriptions that lack fine-grained visual details. However, since our framework is agnostic to the specific use of visual captioning models, we believe that future advances in visual captioning models will mitigate this issue. On the reasoning side, our framework may underperform when the reasoning trace generated by

the LLM is incorrect. However, we view this as a broader limitation of current LLMs, and anticipate that future advances in long-context modeling and reasoning for LLMs will further enhance the performance of our framework.

## Acknowledgements

This work was supported by Laboratory for Analytic Sciences via NC State University, ONR Award N00014-23-1-2356, NIH Award R01HD11107402, and Sony Focused Research award.

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

Our appendix consists of Additional Ablation Study (Section A), More Experimental Results (Section B), Additional Implementation Details (Section C), and Qualitative Results (Section D).

# A  Additional Ablation Study

**Time-aware Caption Representation.** We investigate the effectiveness of integrating time information into captions on the Video-MMMU benchmark. The results are presented in Table 14. The baseline concatenates all captions in temporal order without explicit timestamps, whereas our proposed time-aware method explicitly includes time information. Specifically, before each caption, we add a timestamp indicating the interval from which it was extracted (e.g. 00:00:24 –> 00:00:32: A clear glass bottle being filled with water, surrounded by seashells and a white cloth.) As shown in Table 14, incorporating timestamp information leads to a notable **2.25%** improvement in overall accuracy compared to the baseline. Furthermore, the time-aware method consistently outperforms the baseline across all question categories. These results demonstrate that explicitly providing time information effectively enhances the temporal perception and reasoning capabilities of the LLMs. Consequently, we adopt the time-aware caption representation for all experiments.

| Method | Overall | Perception | Comprehension | Adaptation |
|--------|---------|-----------|---------------|-----------|
| w/o time | 72.86 | 81.00 | 80.67 | 56.90 |
| w/ time | **75.11** | **82.67** | **82.67** | **60.00** |

Table 14: **Time-aware Caption Representation.** Incorporating time information into the captions by adding timestamps depicting time intervals from which the captions were extracted significantly boosts the performance on Video-MMMU.

# B  More Experimental Results

**HourVideo.** We evaluate our method on the development set of HourVideo (Chandrasegaran et al., 2024), a benchmark specifically designed for 3D reasoning over long videos. As shown in Table 15, our method performs surprisingly well, despite not incorporating any explicit 3D modeling. Specifically, it achieves an accuracy of 36.3%, outperforming the concurrent method VAMBA (Ren et al., 2025) by a notable margin of **2.7%**. These results demonstrate that our method is effective in reasoning about both the 3D physical world and hour-long videos.

| Method | Overall Accuracy |
|--------|-----------------|
| Aria | **39.2** |
| Gemini 1.5 Pro | 37.4 |
| Qwen2-VL 7B | 33.8 |
| VAMBA | 33.6 |
| SiLVR (ours) | 36.3 |

Table 15: **Performance of Our Method on HourVideo.** SiLVR outperforms the concurrent work VAMBA by a significant **2.7%**.

**CGBench-Reasoning.** We evaluate our method on the reasoning split of CGBench. As shown in Table16, SiLVR achieves the highest performance among the baseline models. Specifically, SiLVR outperforms Video-R1 by a significant 6.2%. Additionally, SiLVR outperforms the concurrent video reasoning model Video-Thinker-7B by 0.7%, showing the strong video reasoning ability.

**VideoMME Overall Performance.** We evaluate our method on VideoMME and report the overall performance including the short, medium and long splits. As shown in the Table 17, SiLVR achieves strong performance among open-source models. Specifically, SiLVR surpasses the widely used Qwen2.5-VL 7B by 2.6%. In addition, SiLVR also outperforms the video reasoning model Video-R1 by a substantial 12.8%,

| Method | Overall Accuracy |
|---|---|
| Qwen2.5-VL 7B | 32.6 |
| Video-R1 7B | 30.1 |
| Video-Thinker 7B | 35.6 |
| SiLVR (ours) | **36.3** |

Table 16: **Performance of Our Method on CGBench-Reasoning.** SiLVR outperforms the concurrent video reasoning model Video-Thinker-7B by **0.7%**.

highlighting its improved video comprehension capability. While proprietary large models such as Gemini 2.5-Pro achieve higher accuracy, SiLVR remains highly competitive within the open-source models.

| Method | Overall Accuracy |
|---|---|
| *Proprietary* | |
| Gemini 2.5-Flash | 81.5 |
| Gemini 2.5-Pro | 86.9 |
| *Open-source* | |
| Qwen2.5-VL 7B | 71.6 |
| Qwen3-VL 8B | 71.8 |
| Qwen3-VL 32B | 77.3 |
| Video-R1 7B | 61.4 |
| SiLVR (ours) | 74.2 |

Table 17: **Performance of Our Method on VideoMME.** We report the overall accuracy. SiLVR outperforms Qwen2.5-VL 7B by **2.6%** and Video-R1 by **12.8%**, and remains competitive among open-source methods.

**Detailed VideoMME Results.** In Table 18, we present a detailed breakdown of our method's performance on VideoMME. From these results, we observe that our method achieves the lowest accuracy on the Counting Problem. This is likely due to the complexity of counting tasks, which require precise temporal localization of multiple events and subsequent reasoning. Any missed or incorrectly detected events could lead to incorrect answers, making the Counting Problem particularly challenging.

| Question Category | Accuracy |
|---|---|
| Temporal Reasoning (TR) | 74.6 |
| Spatial Reasoning (SR) | 94.6 |
| Action Reasoning (AR) | 76.1 |
| Object Reasoning (OR) | 79.5 |
| Temporal Perception (TP) | 85.5 |
| Spatial Perception (SP) | 74.1 |
| Attribute Perception (AP) | 80.2 |
| Action Recognition (AC) | 68.1 |
| Object Recognition (OC) | 82.5 |
| OCR Problems (OP) | 83.5 |
| Counting Problem (CP) | 50.7 |
| Information Synopsis (IS) | 88.5 |

Table 18: **Detailed Results on VideoMME.** Our method achieves the highest accuracy on Spatial Reasoning while achieving the lowest performance on the challenging Counting Problem.

# C   Additional Implementation Details

## C.1   Captioner

For all LLaVA and Qwen models, we use the prompt "Briefly describe the video within 40 words" to generate captions for each clip. We set `max_new_tokens` to 200 and employ greedy decoding. We utilize the following model variants from Hugging Face: `lmms-lab/llava-onevision-qwen2-7b-ov` for LLaVA-OV 7B, `Qwen/Qwen2.5-VL-7B-Instruct` for Qwen2.5-VL 7B, and `Qwen/Qwen2.5-VL- 72B-Instruct` for Qwen2.5-VL 72B. For the NVILA model, we use the NVILA-8B-Video variant with the prompt "generate caption" to produce captions for each clip. We set `max_new_tokens` to 128 and employ greedy decoding similar to LLaVA and Qwen models.

We use 4 H100 GPUs for generating captions.

## C.2   LLM

We use the default temperature of 1.0 for all LLM experiments. We use the DeepSeek API to run DeepSeek-R1 and DeepSeek-V3 efficiently. To accelerate inference, we implement a parallel processing pipeline with up to 64 concurrent processes, each handling raw captions and subtitles before sending requests to the API. During off-peak hours, this setup allows us to evaluate our method on the complete VideoMME benchmark in under 2 hours at a cost of less than $20. For GPT models, we use the OpenAI API. For Llama-4 models, we use 4×H100 GPUs for local inference. However, we can only process a subset of videos locally due to GPU memory constraints. For long videos, we use Lambda Cloud's API service.

## C.3   Prompt Design

Different VideoQA benchmarks include different types of questions (e.g., multiple-choice, open-ended, or mixed). Additionally, the Grounded VideoQA task in CGBench requires models to predict the temporal boundaries (start/end timestamps) of video segments relevant to each question. To accommodate these differences, we design task-specific prompts. Specifically, for multiple-choice questions, we use the following prompt template:

---

The video's subtitles are listed below.
`Subtitles`.

The video's captions are listed below. Each caption describes a `Clip Length` seconds clip.
`Captions`.

Select the best answer to the following multiple-choice question based on the video and the subtitles. Respond with only the letter (A, B, C, D, E, etc.) of the correct option.

Question: `Question`.
Options: `Options`.

The answer is:

---

For open-ended questions, we use the following prompt template:

> The video's subtitles are listed below.
> `Subtitles`.
>
> The video's captions are listed below. Each caption describes a `Clip Length` seconds clip.
> `Captions`.
>
> Based on the video and the subtitles. Answer the following question with one sentence. Answer the following question based on the video and the subtitles. The answer is short. Please directly respond with the short answer.
>
> Question: `Question`.
>
> The answer is:

For the Grounded VideoQA task, we use the following prompt to generate the start and end seconds of the question-related clips:

> The video's subtitles are listed below.
> `Subtitles`.
>
> The video's captions are listed below. Each caption describes a `Clip Length` seconds clip.
> `Captions`.
>
> Your task is to determine in which intervals the clue exists that contain visual information needed to answer the question.
>
> Question: `Question`.
>
> Only output the answer in the following format:
> [[start1, end1], [start2, end2], ...]
> In this output format, each 'start' and 'end' represents the beginning and end of an interval in seconds (integer) where relevant clues can be found.
> You must provide at least one interval and at most five intervals.
> Here are some example outputs.
> Example 1: [[5, 7]]
> Example 2: [[200, 207], [209, 213], [214, 220]]

## C.4 Evaluation

We evaluate our method on the validation set of MMVU and HourVideo. For the VideoQA task on CGBench, we adopt the `long-acc` setting in which the model takes the entire long video as the input and answers the given questions.

We follow the official evaluation code of each benchmark to make a fair comparison with prior methods. If the official evaluation code is not provided, we use the code from LMMs Eval (Zhang et al., 2024a).

For the Grounded VideoQA task on CGBench, we need to decode the timestamps from the LLM outputs. In practice, we found that the Grounded QA prompt as shown in Section C.3 works surprisingly well, with each output sentence following the list format exactly. Therefore, we simply use the `eval` function in Python to decode the output text into a list object. We then use the official code provided by CGBench to compute the mIoU between the prediction and the ground truth.

## D  Additional Qualitative Results

We present and analyze several of SiLVR's reasoning traces in Figure 6, and Figure 7. From the figures, we observe that SiLVR effectively integrates information from both visual and speech modalities. Furthermore, SiLVR is capable of performing complex video reasoning steps, including step-by-step reasoning, self-correction, self-verification, and using relevant prior knowledge from the LLM to answer a given question.

**Video:** 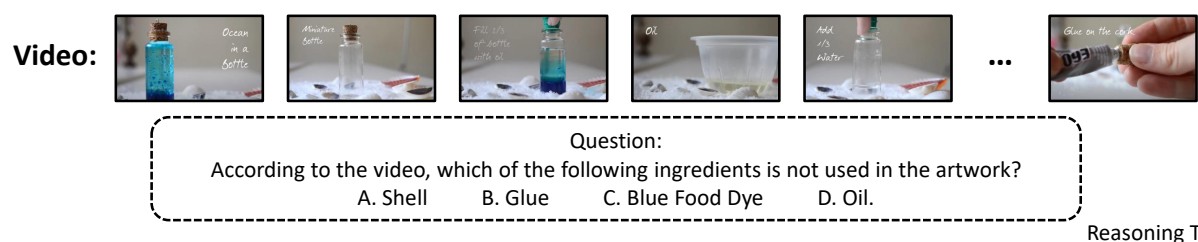

Question:
According to the video, which of the following ingredients is not used in the artwork?
A. Shell     B. Glue     C. Blue Food Dye     D. Oil.

Reasoning Trace

Okay, let's tackle this question. So, the question is asking which ingredient is not used in the artwork based on the video subtitles. The options are shell, glue, blue food dye, oil. Looking at the subtitles provided, let me go through each timestamp to see what's mentioned. Starting from 00:00:00 to 00:00:08, there's a glass bottle with blue liquid and decorative items. The decorative items might include shells, but not sure yet. At 00:00:16, the caption mentions a bottle of E6000 adhesive surrounded by seashells and a plastic cup with yellowish liquid. E6000 is a type of glue, so glue is definitely used (option B is present). Shells are also mentioned here, so shell (A) is used. At 00:00:32, a hand adds a drop of blue food dye to the bottle. That's option C, which is used. Then at 00:00:40, the text says 'Fill 2/3 of bottle with oil' and there's a hand holding a green dropper. So oil (D) is used. Wait, but the question is which is NOT used. Let me check again. All four options are mentioned except one. The options are A: Shell, B: Glue, C: Blue Food Dye, D: Oil. From the subtitles: seashells are mentioned (so A is used). Glue (E6000 adhesive) is present (B used). Blue food dye is added (C used). Oil is instructed to fill 2/3 of the bottle (D used). Wait, all options are used? That can't be….. Maybe the shells are part of the decorative items but not an ingredient used in the artwork's construction? Wait, the question is about ingredients used in the artwork. Are the shells part of the decoration but not part of the main mixture? Let's read the steps again. The steps involve filling the bottle with water (00:00:24), adding blue dye (00:00:32), adding oil (00:00:40). Then there's mention of glue (E6000) which might be for sealing or attaching something. The seashells are surrounding the glue and maybe used as decoration. But perhaps the shells are not part of the liquid mixture inside the bottle but are just part of the surrounding decorations. If the question is about the ingredients inside the bottle, …. Therefore, the shells might be part of the decoration around the bottle in the video but not part of the actual artwork (the bottle's contents). So the answer would be A: Shell, as the shells are not an ingredient used in the artwork. But I need to make sure. The question says \ingredients\, which implies components used in creating the main part of the artwork. If the shells are not inside the bottle or part of its structure, then they are not ingredients. So the answer is A.

Figure 6: **Example 1 of SiLVR's Reasoning Trace.** The question asks which ingredients are not used in the video. Initially, SiLVR identified all listed items as potential ingredients. However, through self-correction, SiLVR correctly recognized that the shell is used as decoration rather than an ingredient.

**Video:** 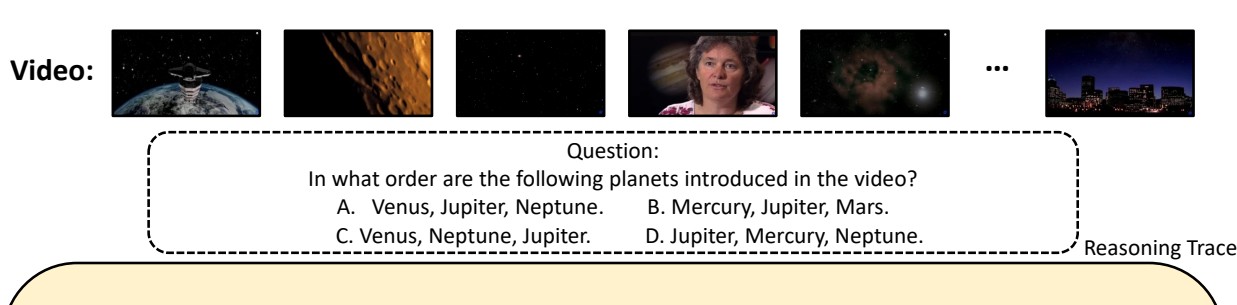

Question:
In what order are the following planets introduced in the video?
A.  Venus, Jupiter, Neptune.     B. Mercury, Jupiter, Mars.
C. Venus, Neptune, Jupiter.     D. Jupiter, Mercury, Neptune.

Reasoning Trace

Okay, let me try to figure this out. The question is asking the order in which certain planets are introduced in the video. The options are A, B, C, D, each listing a sequence of three planets. First, I need to recall the structure of the video based on the provided subtitles and captions. The subtitles start by talking about humanity leaving Earth and exploring the solar system. Then they mention Mercury first, discussing its atmosphere and temperature. The next planet discussed is Venus, with details about its thick atmosphere and the Soviet Venera missions. After Venus, the video moves on to Mars, discussing its potential for human colonization, radiation issues, dust storms, etc. Then the video goes into the outer planets, starting with Jupiter. They talk about its Great Red Spot, atmosphere, and moons like Io. After Jupiter, Saturn is mentioned with its rings and storms. Following Saturn, Neptune is discussed, focusing on its high-speed winds and moon Triton. So the sequence of planets introduced is Mercury → Venus → Mars → Jupiter → Saturn → Neptune. The question asks for the order among Venus, Jupiter, Neptune. From the sequence above, Venus comes before Jupiter, which comes before Neptune. Looking at the options, option A is Venus, Jupiter, Neptune. Option C is Venus, Neptune, Jupiter, which would be incorrect. The other options don't include the correct order. For example, option B lists Mercury, Jupiter, Mars, which isn't correct because Mars comes after Mercury and Venus. So the correct answer should be A.

Figure 7: **Example 2 of SiLVR's Reasoning Trace.** The video sequentially introduces six planets in detail: Mercury, Venus, Mars, Jupiter, Saturn, and Neptune. SiLVR accurately identifies the correct order of the planets and systematically inspects all answer choices, eliminating the incorrect ones through logical reasoning.

