# OpenReview forum: "SiLVR: A Simple Language-based Video Reasoning Framework"
_TMLR — Accepted by TMLR_

### Review · Reviewer_Mgwb · 2025-10-20

**Summary Of Contributions:**

Strengths
1. The model shows significant improvements on multiple benchmarks, especially on Video-MMLU and CGBench, demonstrating strong general reasoning ability across diverse video-language tasks.
2. The proposed SiLVR framework is simple, modular, and effective, and it works well with different reasoning LLMs, showing good generalizability.
3. The introduction of audio information provides complementary cues beyond visual input, effectively improving overall video reasoning performance.

Weaknesses
1. As shown in the experiments, the performance gain mainly comes from the reasoning ability of the LLM (e.g., DeepSeek-R1), rather than the framework itself, raising doubts about SiLVR’s intrinsic contribution.
2. The paper notes that using stronger captioners like Qwen2.5-VL 7B/72B can further boost performance, suggesting the framework’s improvement may rely heavily on external modules.
3. Table 1 lacks comparisons with several important models (e.g., Gemini-2.5 series, Qwen2.5-VL), making the evaluation incomplete.
4. The related work section misses several key prior studies (e.g., MLVU, VideoXL, VideoEspresso, VideoR1), weakening the paper’s contextual grounding.

**Audience:**

Yes

**Audience Explanation:**

Although the proposed approach is not particularly novel in its framework design, the idea of incorporating audio as an additional modality is meaningful and has been under-explored in prior video reasoning research. Moreover, the paper reports remarkably high performance on VideoMMLU (over 80% accuracy), which makes the results interesting to the community and worth understanding in greater depth.

**Broader Impact Concerns:**

No.

**Claims And Evidence:**

No

**Claims Explanation:**

The claims are not fully supported by convincing evidence. The proposed framework essentially functions as an agent-style pipeline rather than a unified end-to-end model. Therefore, directly comparing its performance against fully end-to-end multimodal models is not entirely fair. Moreover, Table 1 omits several highly competitive baselines (e.g., Gemini-2.5 series, Qwen2.5-VL, VideoR1), which further weakens the validity and persuasiveness of the performance claims.

**Requested Changes:**

1. Provide a clearer attribution of performance gains. The paper should better disentangle the contribution of the SiLVR framework itself from the reasoning ability of the underlying LLM (e.g., DeepSeek-R1). This clarification is critical for assessing the originality and true effectiveness of the proposed method.
2. Include more comprehensive model comparisons – Table 1 should incorporate results from additional strong baselines such as Gemini-2.5, Qwen2.5-VL, and VideoR1, which are relevant and frequently cited in the field. This addition is critical to ensure a fair and convincing evaluation.
3. Expand the related work section – The discussion should include important prior works like VisualCoT, MLVU, VideoXL, VideoEspresso, and VideoR1. Their omission weakens the contextual positioning of this paper. This revision is important but not strictly critical.
4. Discuss the role of visual captioners more explicitly – Since the ablation shows that stronger captioners (e.g., Qwen2.5-VL 72B) yield higher accuracy, the paper should clarify to what extent the improvements depend on the chosen captioner rather than the framework itself. This would strengthen the work’s transparency.

---

> ### Author Response · Authors · 2025-11-17
> **Response to Reviewer Mgwb (Part 1/3)**
>
> ### **Weakness 1 & Requested Changes 1. SiLVR’s intrinsic contribution.**
>
> > As shown in the experiments, the performance gain mainly comes from the reasoning ability of the LLM (e.g., DeepSeek-R1), rather than the framework itself, raising doubts about SiLVR’s intrinsic contribution.
>
> > Provide a clearer attribution of performance gains. The paper should better disentangle the contribution of the SiLVR framework itself from the reasoning ability of the underlying LLM (e.g., DeepSeek-R1). This clarification is critical for assessing the originality and true effectiveness of the proposed method.
>
> We thank the reviewer for this observation. We acknowledge that the performance gains are significantly driven by the reasoning capabilities of the LLM. However, we would like to emphasize that **it is SiLVR's framework design that explicitly enables this gain.** By strategically converting long videos into dense language representations, our framework is uniquely positioned to leverage very large and powerful LLMs (e.g., DeepSeek-R1). This stands in sharp contrast to many video-native approaches (e.g., Video-R1), which cannot easily scale to or incorporate such models due to prohibitive GPU memory constraints.
>
> Furthermore, this design makes SiLVR exceptionally modular and flexible. It is a training-free framework that can **"ride the wave" of progress** in foundation models. As new, more powerful video captioners or LLMs are released, they can be integrated seamlessly to instantly improve performance. This modularity also allows for the straightforward incorporation of new modalities and task-specific information (e.g., speech transcripts, object detections) without the complex, resource-intensive training required by other paradigms.
>
> Finally, SiLVR's effectiveness is not tied to one specific captioner or LLM. To demonstrate this **robustness and generality**, we conducted additional experiments on VideoMME. As the results below show, SiLVR benefits from multiple different captioners and LLMs, and it consistently improves with stronger components.
>
> |                          |             |          |          |          |          |
> | ------------------------ | ----------- | -------- | -------- | -------- | -------- |
> | Captioner                | LLM         | Overall  | Short    | Medium   | Long     |
> | NVILA-7B                 | DeepSeek-R1 | 70.3     | 63.2     | 70.4     | 77.3     |
> | Qwen2.5-VL-7B            | DeepSeek-R1 | 70.9     | 63.8     | 72.9     | 76.1     |
> | NVILA-7B + Qwen2.5-VL-7B | DeepSeek-R1 | **74.1** | **68.3** | **74.6** | **79.4** |
>
> |           |                |          |          |          |          |
> | --------- | -------------- | -------- | -------- | -------- | -------- |
> | Captioner | LLM            | Overall  | Short    | Medium   | Long     |
> | NVILA-7B  | DeepSeek-R1    | 70.3     | 63.2     | 70.4     | 77.3     |
> | NVILA-7B  | Gemini-2.5-Pro | 73.1     | 60.3     | **75.3** | **83.8** |
> | NVILA-7B  | GPT-5          | **73.9** | **64.7** | 75.1     | 81.9     |
>
> From these results, we observe that SiLVR supports multiple visual captioners and LLMs, and it consistently benefits from stronger ones. The ensemble captioner (NVILA-7B + Qwen2.5-VL-7B) improves overall accuracy by **3.8%** and **3.2%** over NVILA-7B and Qwen2.5-VL-7B individually. Similarly, using GPT-5 and Gemini-2.5-Pro as LLMs outperform DeepSeek-R1 by **3.6%** and **2.8%** on overall accuracy, respectively. Notably, Gemini-2.5-Pro achieves a **6.6%** improvement on the long split of VideoMME, highlighting that stronger LLMs contribute most significantly to tasks requiring complex temporal reasoning within our framework.
>
> Finally, we wish to position SiLVR as a **powerful alternative and challenging baseline** for current video-native reasoning paradigms (e.g., Video-R1). While end-to-end video-native models are a promising direction, our work demonstrates that a modular, language-centric framework offers a highly competitive, flexible path to complex video understanding. SiLVR's strong performance underscores the viability of this approach, and we view it as a critical baseline that future video-native systems should be measured against. We believe its simple yet effective design offers valuable insights that can help inspire new directions in both language-driven and video-native video understanding research.

---

> ### Author Response · Authors · 2025-11-17
> **Response to Reviewer Mgwb (Part 2/3)**
>
> ### **Weakness 3 & Requested Changes 2. Unfair comparison with baselines.**
>
> > Table 1 lacks comparisons with several important models (e.g., Gemini-2.5 series, Qwen2.5-VL), making the evaluation incomplete.
>
> > The claims are not fully supported by convincing evidence. The proposed framework essentially functions as an agent-style pipeline rather than a unified end-to-end model. Therefore, directly comparing its performance against fully end-to-end multimodal models is not entirely fair. Moreover, Table 1 omits several highly competitive baselines (e.g., Gemini-2.5 series, Qwen2.5-VL, VideoR1), which further weakens the validity and persuasiveness of the performance claims.
>
> > Include more comprehensive model comparisons – Table 1 should incorporate results from additional strong baselines such as Gemini-2.5, Qwen2.5-VL, and VideoR1, which are relevant and frequently cited in the field. This addition is critical to ensure a fair and convincing evaluation.
>
> We thank the reviewer for this crucial suggestion. We agree that a comprehensive comparison against the latest SOTA models is essential to validate SiLVR's performance.
>
> The reviewer correctly notes that SiLVR functions as a modular, agent-style pipeline rather than a unified end-to-end model. We argue this is a deliberate design strength, and a direct comparison is not only fair but necessary to demonstrate that our **modular, training-free** approach can match or exceed the performance of **end-to-end models**.
>
> To provide this comprehensive and transparent evaluation, we have run SiLVR against the baselines that the reviewer suggested (Gemini-2.5 series, Qwen2.5-VL-7B, and Video-R1) on a suite of benchmarks. The results are presented below.
>
> |                  |                            |            |                   |                          |
> | ---------------- | -------------------------- | ---------- | ----------------- | ------------------------ |
> | Model            | Video Reasoning Benchmarks |            |                   | General Video Benchmarks |
> |                  | Video-MMMU                 | Video-MMLU | CGBench-Reasoning | Video-MME                |
> | *Proprietary*      |                            |            |                   |                          |
> | Gemini 2.5-Flash | 79.2                       | -          | -                 | 81.5                     |
> | Gemini 2.5-Pro   | 83.6                       | -          | -                 | 86.9                     |
> | *Open-source*      |                            |            |                   |                          |
> | Qwen2.5-VL 7B    | 50.4                       | 32.9       | 32.6              | 71.6                     |
> | Qwen-2.5-VL 72B  | 61.0                       | 40.5       | -                 | **79.1**                 |
> | Qwen-3-VL 8B     | 72.8                       | -          | -                 | 71.8                     |
> | Qwen-3-VL 32B    | 79.0                       | -          | -                 | 77.3                     |
> | Video-R1         | 52.4                       | -          | 30.1              | 61.4                     |
> | SiLVR (ours)     | **82.7**                   | **83.1**   | **53.4**          | 74.2                     |
>
> These results show that SiLVR is highly effective. On the critical reasoning benchmarks (Video-MMMU, Video-MMLU, CGBench-Reasoning), **SiLVR significantly outperforms every other open-source model**, including the 72B-parameter Qwen2.5-VL and the recent Video-R1. This strongly validates our framework's design.
>
> Furthermore, SiLVR's performance on Video-MMMU (82.7%) is highly competitive with the SOTA proprietary model, Gemini-2.5-Pro (83.6%), demonstrating its capability on complex reasoning tasks.
>
> Finally, we compare our method with other agent-based approaches on VideoMME (long) in the table below.
>
> |              |                                 |
> | ------------ | ------------------------------- |
> | Model        | VideoMME (long) w\.o. subtitles |
> | VideoAgent   | 46.4                            |
> | DrVideo      | 51.7                            |
> | VideoTree    | 54.2                            |
> | VCA          | 56.3                            |
> | SiLVR (ours) | **62.7**                        |
>
> As shown, SiLVR substantially outperforms all other agent-based methods, confirming its superior design for long-form video reasoning.
>
> We will include all this new analysis in the final draft to provide the fair and convincing evaluation the reviewer requested.

---

> ### Author Response · Authors · 2025-11-17
> **Response to Reviewer Mgwb (Part 3/3)**
>
> ### **Weakness 2 & Requested Changes 4. Effects of visual captioners.**
>
> > The paper notes that using stronger captioners like Qwen2.5-VL 7B/72B can further boost performance, suggesting the framework’s improvement may rely heavily on external modules.
>
> > Discuss the role of visual captioners more explicitly – Since the ablation shows that stronger captioners (e.g., Qwen2.5-VL 72B) yield higher accuracy, the paper should clarify to what extent the improvements depend on the chosen captioner rather than the framework itself. This would strengthen the work’s transparency.
>
> We thank the reviewer for this important point, which allows us to clarify a core design principle of our work. We agree that visual captioners play a significant role. In fact, SiLVR is **explicitly designed to be a modular, training-free framework** whose performance scales with the strength of its external modules. We see this not as a limitation, but as its **primary advantage.**
>
> This design intentionally avoids monolithic, end-to-end training. Instead, it creates a flexible framework that can **incorporate rapid advances** in foundation models. When a stronger captioner or a more powerful LLM is released, SiLVR can integrate it seamlessly—without any retraining—to instantly benefit from advances in visual perception or reasoning.
>
> To provide the transparency the reviewer requested, we conducted experiments with a diverse set of components:
>
> 1) **Stronger Modules:** As shown in the table from our previous response, SiLVR's performance consistently improves when integrating a stronger captioner (e.g., the NVILA-7B + Qwen2.5-VL-7B ensemble) or stronger LLMs (e.g., GPT-5 and Gemini-2.5-Pro). This confirms the framework is successfully a) general-purpose and b) capable of leveraging SOTA components.
>
> 2) **Lightweight Modules:** To further demonstrate the framework's robustness, we also conducted a new experiment with a significantly more lightweight captioner, **Qwen-3-3B**.
>
> |               |             |          |          |        |          |
> | ------------- | ----------- | -------- | -------- | ------ | -------- |
> | Captioner     | LLM         | Overall  | Short    | Medium | Long     |
> | Qwen2.5-VL-7B | DeepSeek-R1 | 70.9     | **63.8** | 72.9   | 76.1     |
> | Qwen3-VL-3B   | DeepSeek-R1 | **71.0** | 63.5     | 70.9   | **78.6** |
>
> As this result shows, SiLVR remains effective and achieves a strong result (71.0%) even with a 3B parameter captioner. This demonstrates that the **framework itself is sound** and not exclusively dependent on a specific visual captioner or an LLM.In summary, SiLVR's "reliance" on external modules is a deliberate and core feature. It provides the essential, training-free architecture that effectively converts the rapid, independent progress in captioning and LLMs into direct, immediate gains for complex video reasoning.
>
>
>
> ### **Weakness 4 & Requested Changes 3. Missing prior studies.**
>
> > The related work section misses several key prior studies (e.g., MLVU, VideoXL, VideoEspresso, VideoR1), weakening the paper’s contextual grounding.Expand the related work section – The discussion should include important prior works like VisualCoT, MLVU, VideoXL, VideoEspresso, and VideoR1. Their omission weakens the contextual positioning of this paper. This revision is important but not strictly critical.
>
>
> We thank the reviewer for this helpful suggestion. In our revised manuscript, we will expand the related work section to include a discussion of **MLVU**, **VideoXL**, **VideoEspresso**, and **VisualCoT**. (We would like to gently note that **Video-R1** is already discussed in our current draft, but we will ensure our expanded discussion better integrates it with these other methods.)
>
> This added context will allow us to more clearly differentiate our approach and position SiLVR's unique, training-free, and language-centric framework against these important prior methods, highlighting its distinct advantages.

---

> > ### Comment · Reviewer_Mgwb · 2025-11-18
> >
> > Thank you to the authors for their response and thorough experiments, which have addressed some of the issues I raised. I will increase my expected score.

---

> > > ### Author Response · Authors · 2025-11-29
> > > **Response to Reviewer Mgwb**
> > >
> > > We sincerely thank the reviewer for their time, constructive feedback, and for raising their score. We are pleased that our responses and additional experiments have successfully addressed their concerns.

---

### Review · Reviewer_M43Y · 2025-10-29

**Summary Of Contributions:**

1. This work proposes SiLVR, a training-free, modular, two-stage pipeline: (i) convert video to language via dense clip captions + ASR; (ii) feed the concatenated text to a reasoning LLM (DeepSeek-R1) with an Adaptive Token Reduction (ATR) policy to fit long videos within context.

2. This work reports SOTA or strong results on several long-video QA benchmarks (Video-MME long, Video-MMMU comprehension, Video-MMLU, CGBench, EgoLife) and analyses on VideoMME showing that (a) reasoning LLMs outperform non-reasoning LLMs within this pipeline, (b) speech tokens matter more than caption tokens, and (c) ATR beats fixed clip lengths.

**Additional Comments:**

I want to further encourage the authors to consider the general direction of this paper, which is trying to show that a language-based video reasoning method is a strong baseline. Given the current performance, this seems to be the case. But this is also somewhat superficial. If we truly believe in language-based video reasoning methods, we need to show "why": what are the key factors that an end-to-end vision-only model cannot achieve or would really struggle with? Or if there is no such factor (information-wise, the video should be enough), then how could we make video-only methods stronger, what are the real bottlenecks that limit the current methods?

Also, another direction to think is on scaling up the inference compute, which is clearly what this paper is doing. But there are multiple directions in doing so, for example, simply adding more video frames, or adopting a multi-turn reasoning paradigm, or using different tools, etc. How does this work differ from them? What is the best way to scale up the inference compute and get the most reliable result? If the authors can demonstrate this systematically, that would also be a great contribution to the community.

**Audience:**

Yes

**Audience Explanation:**

While in my opinion, this work merely extends prior works in this line with stronger modules (reasoning LLM, and MLLM as visual captioner), but it can serve as a good literature for people to understand the current progress of video reasoning, and understand the gap between such vision-->text models and vision-only models. In this regard, I believe at least some individuals in TMLR's audience would find this paper useful and interesting.

**Broader Impact Concerns:**

None.

**Claims And Evidence:**

No

**Claims Explanation:**

To be honest, I think the answer above should be ``partially'', but there is no such choice.

Specifically, the core claims (SiLVR > baselines; reasoning LLM benefits; ATR helps) are supported by tables (Table 1,2,7); however, there are fairness and scope gaps that blunt the strength of the conclusions.

The first problem is on the inference efficiency. While Table 3 provides some analysis, it's apparently not enough. As the authors mentioned in the last paragraph of the Related Work, there were many works already that tried to use a visual captioner + an LLM (agent) to perform video reasoning. Doesn't this work fall in this line? Why is there no comparison or discussion with previous works? Even for the compared baselines (DrVideo and VideoTree), there are no discussions of what they are, and what does the performance/efficiency difference mean. In fact, how is comparing this method to a tree-based method meaningful?

In addition, Table 8 clearly shows that video captioners are actually heavy MLLMs, so to be fair, the number of LLM calls should probably also count them as well. The current "1" LLM call is very misleading.

It would also be meaningful to add video-only baseline (and include both without reasoning and with reasoning) to see the performance and efficiency difference, so that we can get a full picture of the compute/performance trade off here and assess whether the proposed method is indeed a good one.

That also comes to the second problem, which is the significance of the contribution. In my opinion, this work is not significantly different from prior works that use an LLM (agent) to process dense visual captions. Such works were also cited by the authors but without any discussion of the differences (last paragraph of the related work section). Does this work simply use a stronger visual captioner and a stronger reasoning LLM? What's the key difference? What's the real contribution here? The authors should make this clear!

**Requested Changes:**

I have mentioned most of my concerns and requested changes above.

Here is an additional one: More analysis in the ablation studies. For example, in Table 9, why is DeepSeekR1 the strongest? In most other benchmarks (e.g. LMArena), GPT-4.1 is considerably better than DeepSeekR1, why here they are comparable? Is it because this task is different or the benchmark is more suitable for DeepSeekR1? Similarly, other ablations also lack in-depth analysis.

---

> ### Author Response · Authors · 2025-11-17
> **Response to Reviewer M43Y (Part 1/3)**
>
> ### **Weakness 1. More details about the efficiency analysis.**
>
> > The first problem is on the inference efficiency. While Table 3 provides some analysis, it's apparently not enough. Even for the compared baselines (DrVideo and VideoTree), there are no discussions of what they are, and what does the performance/efficiency difference mean. In fact, how is comparing this method to a tree-based method meaningful?
>
> We thank the reviewer for this detailed feedback. We agree that a more in-depth discussion of these baselines and the _nature_ of the efficiency comparison is essential to clarify our contributions. We will add the following discussion to the final draft.
>
> **1. Why the Comparison to Tree-Based Methods is Meaningful**
>
> We apologize for the lack of context. The comparison to `VideoTree` and `DrVideo` is highly meaningful because **all three are 'agent-style' frameworks that solve the long-video problem using a similar high-level pipeline: 1) compress the video into a language representation, and 2) apply an LLM for reasoning.**
>
> The key difference—and the core of our comparison—lies in their _strategy_ for doing so:
>
> - `VideoTree`: Employs a _complex hierarchical search_. It builds a tree of video segments using visual clustering and relevance scoring, adaptively pruning the video to focus on important frames.
>
> - `DrVideo`: Uses an _iterative refinement_ strategy. It decomposes the video, captions clips, and then performs multi-round LLM reasoning to retrieve evidence and refine its predictions.
>
> - `SiLVR` **(Our Approach)**: Employs a _simple, single-pass modular framework_. It avoids complex hierarchical search or iterative refinement, making it simpler and more scalable by design.
>
> **2. What the Performance/Efficiency Differences Mean**
>
> To clarify the efficiency metrics in Table 3, all methods were benchmarked under identical conditions (A6000 GPU, VideoMME-Long dataset). We measured the average processing time per video, broken down into the two stages shared by all methods: 1) Visual Captioning and 2) LLM Reasoning.
>
> The results show that SiLVR's simpler single-pass design achieves a superior performance-efficiency trade-off, demonstrating both higher accuracy and better overall efficiency than the more complex `VideoTree` and `DrVideo` methods.
>
> **3. Direct Comparison to VideoTree's Efficiency Strategy (SiLVR-fast)**
>
> The reviewer correctly notes that `VideoTree`'s core innovation for efficiency is its tree-based _pruning_ of the video to reason over key segments. To create the most direct and fair comparison against this pruning strategy, we introduced **SiLVR-fast**.
>
> This variant uses a much simpler pruning strategy: a larger clip length. This reduces the number of VLM calls and the number of generated captions. The results in Table 3 are therefore highly significant: **SiLVR-fast outperforms VideoTree in both accuracy (+3.0%) and total runtime (2.0x faster).**
>
> This suggests that our simple, single-pass framework is not only more effective but also more efficient than more complex hierarchical search methods. We will add this complete discussion to the paper to make the rationale and results of our efficiency analysis clear.
>
>
> ### **Weakness 2. Misleading LLM calls.**
>
> > In addition, Table 8 clearly shows that video captioners are actually heavy MLLMs, so to be fair, the number of LLM calls should probably also count them as well. The current "1" LLM call is very misleading.
>
> We thank the reviewer for highlighting this ambiguity. We agree that the "1 LLM call" entry in the table is misleading when presented without full context, and we will revise our analysis to be more transparent.
>
> The reviewer is correct: the visual captioners (e.g., NVILA-7B) are themselves MLLMs. Our framework makes multiple calls to these **VLMs (Visual Language Models)** to process video segments.
>
> Our intention with the "1 LLM call" metric was to _specifically_ measure the call to the final, heavyweight **Reasoning LLM** (e.g., DeepSeek-R1, 672B). We isolated this metric because:
>
> 1. **Vast Scale Difference:** The reasoning LLM we use is nearly **100 times larger** than the VLM captioner. The cost of this single reasoning call is a primary computational bottleneck and a key point of comparison.
>
> 2. **Baseline Parity:** The number of _VLM captioning calls_ in our method is designed to be comparable to other agent-based methods like `VideoTree`.
>
> In the revised draft, we will replace this ambiguous metric. We will make the analysis in Table 3 (and Table 8) clearer by explicitly breaking down the costs into:
>
> 1. **VLM Calls (Captioning)**: The number of calls to the VLM.
>
> 2. **Reasoning LLM Calls**: The number of calls to the heavyweight LLM.

---

> ### Author Response · Authors · 2025-11-17
> **Response to Reviewer M43Y (Part 2/3)**
>
> ### **Weakness 3. Comparison to Video-only baselines**
>
> > It would also be meaningful to add video-only baseline (and include both without reasoning and with reasoning) to see the performance and efficiency difference, so that we can get a full picture of the compute/performance trade off here and assess whether the proposed method is indeed a good one.
>
> We thank the reviewer for this suggestion. In the table below, we compare our method with Qwen2.5-VL-7B and Video-R1. To ensure fairness, we run all methods on a single A6000 GPU and measure the average processing time per video on the Video-MME (Long) dataset. For the Qwen2.5-VL-7B (768 frames) setting, we use four A6000 GPUs due to limited memory.
>
> |                           |                   |          |
> | ------------------------- | ----------------- | -------- |
> | Method                    | Total Runtime (s) | Accuracy |
> | SiLVR-fast (ours)         | 83                | 57.2     |
> | SiLVR-best (ours)         | 442               | **62.7** |
> | Qwen2.5VL-7B (32 frames)  | 5.2               | 44.7     |
> | Qwen2.5VL-7B (768 frames) | 115.8             | 50.8     |
> | Video-R1                  | 14.2              | 50.2     |
>
> These results lead to two main observations:
>
> 1. **Performance:** Both SiLVR configurations significantly outperform the video-native end-to-end models (i.e., Qwen-2.5-VL and Video-R1). Our **SiLVR-best (62.7%)** achieves a **+12.5%** absolute gain over Video-R1 (50.2%), a **+18.0%** gain over Qwen2.5-VL-7B 32 frames (44.7%), and a **+11.9%** gain over Qwen2.5-VL-7B 768 frames (50.8%), .
>
> 2. **Compute/Performance Trade-off:** Our **SiLVR-fast** configuration not only **surpasses the accuracy** of both SOTA video-native models (e.g., +6.4% over Qwen2.5-VL-7B 768 frames) but also **does so at a much lower computational cost** (83s total runtime).
>
> This suggests that our language-driven video reasoning approach is **both significantly more accurate _and_ more efficient** than current end-to-end video-native reasoning models.
>
> We will add this analysis and table to the final draft to fully address this point.
>
>
>
> ### **Requested Changes 1. More analysis in the ablation studies**
>
> > Here is an additional one: More analysis in the ablation studies. For example, in Table 9, why is DeepSeekR1 the strongest? In most other benchmarks (e.g. LMArena), GPT-4.1 is considerably better than DeepSeekR1, why here they are comparable? Is it because this task is different or the benchmark is more suitable for DeepSeekR1? Similarly, other ablations also lack in-depth analysis.
>
> We note that the performance of GPT-4.1 and DeepSeek-R1 is comparable. For example, in LMArena (Text Arena), DeepSeek-R1 slightly outperforms GPT-4.1 in the overall score (1418 vs. 1412). This is consistent with our findings that DeepSeek slightly outperforms GPT 4.1 (by 0.8%) on VideoMME. We will include this clarification in the paper.
>
> We will also expand the discussion for all ablation sections. We will go beyond just reporting the numbers and provide a more in-depth discussion, similar to the one above, explaining _what_ the results mean and _why_ a particular component (like our Adaptive Token Reduction) is or is not effective, and how it contributes to the overall framework.

---

> ### Author Response · Authors · 2025-11-17
> **Response to Reviewer M43Y (Part 3/3)**
>
> ### **Weakness 4. Technical Novelty.**
>
> > As the authors mentioned in the last paragraph of the Related Work, there were many works already that tried to use a visual captioner + an LLM (agent) to perform video reasoning. Doesn't this work fall in this line? Why is there no comparison or discussion with previous works?
>
> > That also comes to the second problem, which is the significance of the contribution. In my opinion, this work is not significantly different from prior works that use an LLM (agent) to process dense visual captions. Such works were also cited by the authors but without any discussion of the differences (last paragraph of the related work section). Does this work simply use a stronger visual captioner and a stronger reasoning LLM? What's the key difference? What's the real contribution here? The authors should make this clear!
>
> Prior works (like those in the last paragraph of our related work) almost exclusively focus on a _captions-only_ pipeline for general video understanding/perception tasks. SiLVR is designed for **complex reasoning**, which requires integrating information from **multiple modalities**. Our key difference is the simple and effective integration of both visual (dense captions) and audio (ASR transcripts) modalities into a unified language representation. This makes our framework fundamentally more powerful for reasoning about real-world videos where speech is critical. A _direct consequence_ of integrating both dense captions and ASR from long videos is a significant increase in context length. Our proposed **Adaptive Token Reduction (ATR)** addresses this problem by fitting rich, dual-modality information within an LLM's context window while preserving the most salient information for the reasoning task.
>
> In summary, the key differences compared to prior text-based video approaches are as follows:
>
> - **Prior Simple Agents (e.g.,** `VideoAgent`**)**: Process short videos, use captions-only, and have no long-context management.
>
> - **Hierarchical Agents (e.g.,** `VideoTree`**,** `DrVideo`**)**: Introduce high complexity (hierarchical trees, iterative search) to handle long videos, but are still captions-only and computationally expensive.
>
> - **SiLVR (Ours)**: Employs a **simple single-pass** design, is **multimodal (visual + audio)**, and uses **ATR** to efficiently manage long context.
>
> We would also like to note that SiLVR's effectiveness is not tied to one specific captioner or LLM. To demonstrate this **robustness and generality**, we conducted additional experiments on VideoMME with multiple different captioners and LLMs (see below).
>
>
> |                          |             |          |          |          |          |
> | ------------------------ | ----------- | -------- | -------- | -------- | -------- |
> | Captioner                | LLM         | Overall  | Short    | Medium   | Long     |
> | NVILA-7B                 | DeepSeek-R1 | 70.3     | 63.2     | 70.4     | 77.3     |
> | Qwen-2.5-7B              | DeepSeek-R1 | 70.9     | 63.8     | 72.9     | 76.1     |
> | NVILA-7B + Qwen2.5-VL-7B | DeepSeek-R1 | **74.1** | **68.3** | **74.6** | **79.4** |
> | Qwen3-VL-3B              | DeepSeek-R1 | 71.0     | 63.5     | 70.9     | 78.6     |
>
> |           |                |          |          |          |          |
> | --------- | -------------- | -------- | -------- | -------- | -------- |
> | Captioner | LLM            | Overall  | Short    | Medium   | Long     |
> | NVILA-7B  | DeepSeek-R1    | 70.3     | 63.2     | 70.4     | 77.3     |
> | NVILA-7B  | Gemini-2.5-Pro | 73.1     | 60.3     | **75.3** | **83.8** |
> | NVILA-7B  | GPT-5          | **73.9** | **64.7** | 75.1     | 81.9     |
>
> From these results, we observe that SiLVR supports multiple visual captioners and LLMs, and it consistently benefits from stronger ones. The ensemble captioner (NVILA-7B + Qwen2.5-VL-7B) improves overall accuracy by **3.8%** and **3.2%** over NVILA-7B and Qwen2.5-VL-7B individually. Additionally, SiLVR remains effective and achieves a strong result (70.1%) even with a 3B parameter captioner. Similarly, using GPT-5 and Gemini-2.5-Pro as LLMs outperform DeepSeek-R1 by **3.6%** and **2.8%** on overall accuracy, respectively. Notably, Gemini-2.5-Pro achieves a **6.6%** improvement on the long split of VideoMME, highlighting that stronger LLMs contribute most significantly to tasks requiring complex reasoning within our framework.
>
> Finally, we would like to position SiLVR as a **powerful alternative and challenging baseline** for current video-native reasoning paradigms (e.g., Video-R1). By showing that a simple, modular, language-centric approach can outperform complex end-to-end models, we hope to inspire new directions in both language-driven and video-native multimodal reasoning research.

---

> > ### Comment · Reviewer_M43Y · 2025-11-28
> >
> > I would like to thank the authors for their comprehensive replies, additional experiments and explanations, and the revised manuscript.
> >
> > Most of my concerns were addressed. However, regarding the technical novelty, I am still a bit stuck here:
> > 1. The main difference between this work and previous similar works is that this work uses more than dense captions --- it also uses ASR
> > 2. And because dense captions + ASR require a lot of context, this work proposes ATR
> >
> > However, I think the first novelty is very limited, it's more like a naive baseline that other works should include. Dense captions have been used for a while. For ASR, some works also have included them (for example, MINERVA Tab.1, Tab.2 have shown very comprehensive results regarding ASR).
> >
> > That leaves the core contribution to be the ATR, which is still good, but apparently lacks enough comparison with other methods and ablation analysis --- There should be many other ways to deal with long context! The ATR is itself a somewhat simple method (which is good), but it needs more comparison with alternatives if it's the major contribution.
> >
> > Therefore, overall I am still not convinced with the significance of the technical novelty of this work.

---

> > > ### Author Response · Authors · 2025-11-29
> > > **Response to Reviewer M43Y**
> > >
> > > We sincerely appreciate the reviewer’s continued engagement and thoughtful follow-up, as well as the acknowledgment that most of their previous concerns have been addressed.
> > >
> > > Regarding the remaining hesitation about technical novelty, we would like to gently contextualize our contribution within TMLR’s specific acceptance criteria. The TMLR guidelines explicitly state:
> > >
> > > > “Crucially, \[the criteria] should not be used as a reason to reject work... Nor should it form the basis for rejecting work on a method considered not ‘novel enough’, as novelty of the studied method is not a necessary criteria for acceptance. We explicitly avoid these terms (‘significant’, ‘impactful’, ‘novel’), and focus instead on the notion of ‘interest’.”
> > >
> > > In alignment with this principle, our work intentionally prioritizes simplicity and effectiveness over methodological novelty. We make this clear in the manuscript (Page 2, Paragraph 3):
> > >
> > > > “While SiLVR is not based on any new complex design choices, it is simple, modular, training-free, and highly performant and generalizes to multiple complex video-language understanding tasks.”
> > >
> > > The reviewer correctly observes that the integration of dense captions and ASR, managed by the Adaptive Context Reduction (ATR) scheme, is conceptually straightforward. The reviewer notes this seems like a "naive baseline that other works should include." We agree with this assessment, and our core contribution is the **empirical demonstration of its surprising effectiveness**.
> > >
> > > Our work rigorously validates the intuition that this straightforward framework achieves remarkably strong results across diverse video reasoning benchmarks. We believe this provides a crucial, generalizable insight for the TMLR audience: highly complex architectural innovations may not be necessary to advance performance in video reasoning.
> > >
> > > By establishing how powerful this accessible, training-free approach is, we provide a necessary foundation for the community, fulfilling our stated goal (Page 2/3):
> > >
> > > > "We believe the simple yet effective design of SiLVR will enable the research community to build on our work and use our simple framework as a baseline to develop even more powerful video-language reasoning models."
> > >
> > > We hope this clarifies the nature of our contribution within the context of TMLR’s guidelines, and we thank the reviewer again for their insightful comments, which have significantly strengthened the manuscript.

---

> > > > ### Comment · Reviewer_M43Y · 2025-11-29
> > > >
> > > > I’d like to thank the authors for their reply. It’s quite a strong rebuttal.
> > > >
> > > > But I think it might conflate two things: simplicity and lack of clarity in terms of contribution.
> > > >
> > > > As the author agreed before: the paradigm of using a visual captioner + an LLM (agent) is not new, so this framework can hardly be claimed as the contribution of this particular work. Rather, the main novelty lies in using an ASR and using ATR to manage context. But like I mentioned before, if these are the main contributions, then there lack enough experiments to study their effectiveness and variants.
> > > >
> > > > Also, I just found that the works in the last part of the related work are still not being discussed?
> > > >
> > > >
> > > >   Another line of work explores training-free frameworks that first convert raw
> > > > videos into dense visual captions, then perform reasoning with off-the-shelf LLMs (Zhang et al., 2023a; Wang
> > > > et al., 2024b; Fan et al., 2024; Ma et al., 2024; Liao et al., 2024; Wang et al., 2024e; Min et al., 2024)
> > > >
> > > > Maybe it’s due to space limitations. But then I would think: will any of this previous work already be good enough if they use a reasoning LLM? And maybe just add ASR on top of it? These lead still quite some room for doubts.
> > > >
> > > > Overall, I would think this is more like a request for clarity of the contributions: is it introducing the reasoning LLM? Is it using ASR and ATR?
> > > > I might miss something. So feel free to point out to me.
> > > >
> > > > Best regards

---

> > > > > ### Author Response · Authors · 2025-11-29
> > > > > **Response to Reviewer M43Y (Part 1/2)**
> > > > >
> > > > > We thank the reviewer for the engaging discussion and the opportunity to further clarify the contributions of our work. We appreciate the acknowledgement that our previous reply constituted a strong rebuttal.
> > > > >
> > > > > We understand the reviewer's hesitation regarding the clarity of the contribution when viewed through the lens of traditional methodological novelty. However, we wish to frame our contributions according to TMLR’s acceptance criteria, which explicitly prioritize "interest" and "generalizable insights" over novelty:
> > > > >
> > > > > > “Nor should it form the basis for rejecting work on a method considered not ‘novel enough’... If the authors make it clear that there is something to be learned by some researchers in their area from their work, then the criterion of interest is considered satisfied.”
> > > > > >
> > > > >
> > > > > The reviewer correctly observes that the paradigm of using a visual captioner + LLM is not new. Our contribution is not the invention of this paradigm, but rather its advancement and the **empirical demonstration of its surprising effectiveness for complex video reasoning**—a domain where prior language-based approaches have been limited.
> > > > >
> > > > > We address the reviewer's specific points below to clarify our core contributions and their interest to the TMLR audience.
> > > > >
> > > > > ### **1. Distinction from Prior Language-Based Frameworks**
> > > > >
> > > > > The reviewer requested further discussion on the distinction between SiLVR and the related training-free frameworks cited in our paper (Zhang et al., 2023a; Wang et al., 2024b; Fan et al., 2024; Ma et al., 2024; Liao et al., 2024; Wang et al., 2024e; Min et al., 2024). We apologize if the discussion in the Related Work (Page 3) was not sufficiently clear due to space constraints.
> > > > >
> > > > > While these works also use a Captioner+LLM approach, they differ significantly from SiLVR:
> > > > >
> > > > > - **Modality and Focus:** Most prior works are **captions-only** and primarily focused on general perception rather than complex reasoning. SiLVR explicitly integrates dense visual captions and ASR transcripts, which is crucial for real-world video reasoning (see evidence in Point 2 below).
> > > > > - **Complexity and Efficiency:** To handle long videos, several recent methods (e.g., VideoTree (Wang et al.), DrVideo (Ma et al.)) employ complex, **iterative or hierarchical** agent designs (e.g., hierarchical trees, iterative search/retrieval). These are computationally expensive and require multiple LLM calls. In contrast, SiLVR employs a **simple, single-pass design**.
> > > > >
> > > > > **Empirical Comparison:** Crucially, we provide direct comparisons against these methods in **Table 3 and Table 4**. SiLVR substantially outperforms all compared agent-based methods (VideoAgent, DrVideo, VideoTree, VCA) on Video-MME (long). Furthermore, Table 4 demonstrates that SiLVR is significantly more efficient (e.g., SiLVR-fast is 2.0x faster than VideoTree and 5.3x faster than DrVideo) while achieving higher accuracy.
> > > > >
> > > > > The reviewer asks: *"Will any of this previous work already be good enough if they use a reasoning LLM? And maybe just add ASR on top of it?"*
> > > > >
> > > > > Our analysis suggests the answer is no. Simply adding dense ASR to existing frameworks (like VideoTree or DrVideo) would overwhelm their context management strategies, which are already complex. SiLVR’s contribution lies in the effective synthesis of these elements (Reasoning LLM + ASR + Dense Captions) managed efficiently by Adaptive Context Reduction (ACR).

---

> > > > > ### Author Response · Authors · 2025-11-29
> > > > > **Response to Reviewer M43Y (Part 2/2)**
> > > > >
> > > > > ### **2. Validation of ASR and ACR (ATR)**
> > > > >
> > > > > The reviewer noted: *"...the main novelty lies in using an ASR and using ATR to manage context. But like I mentioned before, if these are the main contributions, then there lack enough experiments to study their effectiveness and variants."*
> > > > >
> > > > > We respectfully highlight that we have conducted extensive experiments to validate the effectiveness and explore variants of both ASR and ACR (Adaptive Context Reduction, referred to as ATR by the reviewer) in Section 4.7 (Ablation Studies) and Section 4.5 (Efficiency Analysis) as well as additional experiments in our rebuttal.
> > > > >
> > > > > - **Validation of ASR Effectiveness:**
> > > > >     - **Table 6** (Impact of ASR) demonstrates that incorporating ASR provides substantial gains across all eight benchmarks (e.g., +12.0% on Video-MMMU, +15.0% on Video-MME).
> > > > >     - **Table 7** (Token Reduction Analysis) analyzes the relative importance of speech vs. visual tokens.
> > > > > - **Validation of ACR (ATR) Effectiveness and Variants:**
> > > > >     - **Table 8** (ACR vs. fixed clip length) shows that ACR outperforms the best fixed-length baseline (the standard approach) by 2.5%, demonstrating its effectiveness in balancing context length and information granularity.
> > > > >     - **Tables 9 and 10** explore alternative strategies (changing word limits, non-uniform sampling).
> > > > >
> > > > > We believe these experiments provide a thorough analysis of the design choices for ASR and ACR within our framework.
> > > > >
> > > > > ### **Summary and Interest to the Community**
> > > > >
> > > > > In alignment with TMLR’s criteria, we believe that our work will generate significant "interest" for the video understanding research community. SiLVR is the first work to rigorously demonstrate that a simple, modular, multimodal language-based framework can achieve state-of-the-art results on complex video reasoning tasks.
> > > > >
> > > > > This offers a crucial, generalizable insight for the research community: highly complex architectural innovations may not be necessary to advance the state-of-the-art in video reasoning.
> > > > >
> > > > > Our comprehensive evaluation—spanning video QA, knowledge acquisition (Table 5), temporally grounded QA (Table 5), and detailed reasoning analysis (Section 4.3)—provides valuable empirical data. By establishing this strong, accessible baseline, we position SiLVR as a powerful alternative to video-native reasoning paradigms (e.g., Video-R1), satisfying the criterion that "there is something to be learned by some researchers in their area."

---

> ### Comment · Reviewer_M43Y · 2025-11-29
>
> Thanks again for the comprehensive reply from the authors. The experiment results are very supportive of the arguments, I’m sorry for not checking them in depth before (it’s been quite a while from my first read).
>
> However, I want to respectfully say that it’s still obvious that if the main contribution is ATR (to make previous dense captioning methods work), this paper would have to be written completely differently —- it’s not just about effectiveness of ASR or ACR. It’s more about how one would sell it, analyze it, and compare with other methods/baselines. But I already feel the authors are seeing me as being unreasonable. So I don’t think it’s meaningful to further argue about that.
>
> I don’t have anything against the paper from being accepted now. I would leave the Action Editors to decide.

---

### Review · Reviewer_d1wR · 2025-11-03

**Summary Of Contributions:**

This paper introduces a simple multi-stage video reasoning framework that relies on (1) extracting multi-modal information as text (e.g., audio -> ASR transcription, visual -> captions) and then (2) performing complex reasoning on text-only information. This framework builds upon similar prior work, and is different from (single-stage) multi-modal LLMs. The authors also rely on an adaptive token reduction strategy to maximize the context utilization of the LLM for any given video, by iteratively finding the proper segment size for captioning. The proposed method is applied on multiple video VQA benchmarks including Video-MME, Video-MMMU, Video-MMLU, CGBench,
and EgoLife, comparing with prior art (both proprietary and open-source) and showing meaningful gains. The ablation study further provides insights into the impact of reasoning, context utilization, and different modules of the pipeline.

**Additional Comments:**

N/A

**Audience:**

Yes

**Audience Explanation:**

Multi-modal learning:
As this paper provides a simple strategy to process multi-modal information as text, it would be useful to the community working on multi-modal learning. Particularly, since the pipeline is training-free and relies on all open-sourced components, and the authors pledge to release their code, the community will be able to easily adopt and build on top of it.

Video reasoning:
Similarly, the community working on video reasoning will benefit from this work. It presents a non-trivial improvement on video VQA benchmarks with varying complexity, and the insights provided will be useful for the next generation of models.

**Broader Impact Concerns:**

Since the proposed method is training-free, any limitations or biases in pretrained models may still exist in the outputs. However, there are no obvious ethical concerns specific to this method, other than the general ones that exist for any LLM-based content understanding approach.

**Claims And Evidence:**

Yes

**Claims Explanation:**

Clear technical formulation:
This paper is well-motivated and is supported by a clear technical formulation. Relying on dedicated modules for visual and audio signal extraction (i.e., conversion to text) and then processing with a specialized text reasoning model makes sense. Also, the idea to feed the maximum amount of information with adaptive token reduction is novel and useful. The whole formulation is presented clearly with pipeline diagrams and example figures, making it easier for the reader to understand.

SOTA performance:
This achieves SOTA performance among proprietary and open-source video reasoning models on video VQA benchmarks with varying complexity. Such a performance highlights the utility of the presented claims.

Ablation studies:
The authors provide ablation studies to validate almost all of the design decisions, analysing each component of the pipeline. Moreover, they also present interesting insights into how the proposed method improves on particular dimensions such as reasoning and context utilization.

Qualitative results:
The provided qualitative results in the form of reasoning traces are quite revealing and provide meaningful information on how the multi-modal inputs are used to correct the initially wrong responses.

**Requested Changes:**

**Critical for recommendation:**

Although the proposed token reduction is a novel idea, I wonder if this is the optimal way to do this. There are multiple other options I would consider as relevant ablations:
- What happens if we change the word limit of each segment caption, rather than changing the number of segments
- In the current iteration, each segment has equal importance (similar to uniform sampling). However, many works have shown that not all regions/segments are equally important. Is there a way to utilize such non-uniform sampling to better utilize the information fed within the context length? In this context, VideoAgent, LVNet, VideoTree are some literature that should be in the comparison, at least in terms of the technical differences.
- Segments used may have weird breaks in the current version. Does it make sense to consider more semantic segments, rather than uniform sampling?

Taking a step back, is utilizing the context length to the maximum, the best thing to do? This is not always the case, and in fact, there is evidence to the contrary (i.e., even within the context length, the longer inputs results in reduced performance due to attention dilution. For instance, check "Same task, more tokens: the impact of input length on the reasoning performance of large language models" and "Language Repository for Long Video Understanding". This discussion should be taken into account to validate the assumption here, that maximizing the context length is better.

Also, the term "Adaptive Token Reduction" is misleading here. This is different from token reduction methods such as token merging, token pruning, and token learning. This is rather "context reduction" in text space rather than in token space. I would ask the authors to reconsider the naming here for better clarity.

I see that the proprietary models considered here are somewhat outdated. I understand the field is moving really fast, and it's hard to always keep up. But, it would be nice to see at least some minimal comparisons with most up-to-date models (e.g., Gemini-2.5, Qwen3-VL).

In the experiment in Table 2, I wonder if the comparison between Llama4 and Deepseek-R1 is the best way to show the impact of reasoning capability, because there are many differences between these two models. To make a better comparison and isolate the reasoning capability here, does it make sense to use a comparable non-reasoning model (not sure with Deepseek-V3 should be the comparison here-- Table 9 shows it is fairly-matched with R1, or using a model that supports turning off reasoning tokens)? Otherwise, the claim here is not well-supported.

**Strengthen the work:**

Although the proposed multi-stage pipeline provides more-flexibility to use dedicated models (captioner, ASR, reasoner), it may also suffer from loss of information (particularly, in vision -> text) compared to a multi-modal single-stage model. I would like more discussion on this to be included in the paper. The authors also highlight the benefit of “Performing reasoning entirely in language space”. Is this inherently better than natively multi-modal reasoning?

How is ASR segmented (sentence-level, ...)? Please provide more details.

Why do the reported results only include comprehension split (Video-MMMU) and long-split (Video-MME)?

Please report LLM model sizes in all tables, as it gives more context when comparing different methods.

Table 6: Speech is definitely more semantic/compressed than visual captions, and contains high-density ground-truth information compared to generated noisy information. This may be discussed with the results in this table. Also, please report the context length used in each setting in this table to highlight that it is a fair comparison.

---

> ### Author Response · Authors · 2025-11-17
> **Response to Reviewer d1wR (Part 1/5)**
>
> ### **Requested Changes 1. More ablations on Adaptive Token Reduction**
>
> > Although the proposed token reduction is a novel idea, I wonder if this is the optimal way to do this. There are multiple other options I would consider as relevant ablations:* What happens if we change the word limit of each segment caption, rather than changing the number of segments
>
> > In the current iteration, each segment has equal importance (similar to uniform sampling). However, many works have shown that not all regions/segments are equally important. Is there a way to utilize such non-uniform sampling to better utilize the information fed within the context length? In this context, VideoAgent, LVNet, VideoTree are some literature that should be in the comparison, at least in terms of the technical differences.
>
> > Segments used may have weird breaks in the current version. Does it make sense to consider more semantic segments, rather than uniform sampling?
>
> We thank the reviewer for these constructive suggestions. To address reviewer’s concerns, we ran new experiments ablating alternative designs for Adaptive Token Reduction.
>
> **Changing captioner word limits.**
>
> We modified the captioning prompt to constrain the output length (e.g., “Describe the video within 40 words” (default), “Describe the video within 20 words,” “Describe the video within 10 words”). When the context limit was exceeded, we progressively reduced the per-segment word cap (40 words → 20 words → 10 words). The results on VideoMME are shown below.
>
> |                       |                    |
> | --------------------- | ------------------ |
> | Variant               | Accuracy (Overall) |
> | Change Segment Number | 76.7               |
> | Change Word Limit     | 76.6               |
>
> From the results, we can see that changing word limit yields similar results to changing the segment number. We will include this discussion in the final draft.
>
> **Comparison between uniform and non-uniform sampling.**
>
> We experimented with the non-uniform frame sampling strategy used in VideoTree and compared it against uniform sampling in our framework. Specifically, VideoTree extracts CLIP features for each frame, performs k-means clustering into 32 groups, and selects the cluster centers as representative frames. We used these non-uniformly sampled frames in our framework and compared them to uniform sampling on VideoMME (Long). The results are as follows:
>
> |                 |             |
> | --------------- | ----------- |
> | Sampling Method | Acc. (Long) |
> | Non-Uniform     | 77.6        |
> | Uniform         | 76.8        |
>
> From the table, we can observe that **SiLVR is not tied to uniform sampling** and can **immediately benefit from more advanced, non-uniform sampling techniques.** The +0.8% improvement confirms this is a valuable enhancement, and we will add this discussion to the final draft.
>
> **Semantic segments rather than uniform sampling.**
>
> We thank the reviewer for the valuable suggestion. Similar to non-uniform sampling, semantic segmentation is a natural extension of our framework, and it can further enhance segment quality and thus improve our framework. While a full implementation of a temporal segmentation pipeline is non-trivial and beyond the scope of this rebuttal, we see this as a natural and promising extension of our work. We will add this to our future work section, as it builds directly on the modular strengths of the SiLVR framework.

---

> ### Author Response · Authors · 2025-11-17
> **Response to Reviewer d1wR (Part 2/5)**
>
> ### **Requested Changes 2. Effects of maximum LLM context length**
>
> > Taking a step back, is utilizing the context length to the maximum, the best thing to do? This is not always the case, and in fact, there is evidence to the contrary (i.e., even within the context length, the longer inputs results in reduced performance due to attention dilution. For instance, check "Same task, more tokens: the impact of input length on the reasoning performance of large language models" and "Language Repository for Long Video Understanding". This discussion should be taken into account to validate the assumption here, that maximizing the context length is better.
>
>
> We thank the reviewer for this insightful observation. We agree that there might be a performance trade-off between the amount of visual information and the context length in our framework. Specifically, increasing the number of sampled clips yields more fine-grained visual information but also increases the context length which might degrade LLM’s reasoning performance. We conducted an experiment to analyze this exact trade-off. We varied the clip length to control the number of generated captions, measured the resulting average context length, and evaluated the overall accuracy on VideoMME. The results are shown in the table below.
>
> |                        |                |
> | ---------------------- | -------------- |
> | Average Context Length | Acc. (Overall) |
> | 9.3k                   | 70.3           |
> | 18.2k                  | 74.2           |
> | 29.5k                  | 76.7           |
>
> From the table, we can see that the performance consistently improves as context length increases. This result suggests that the benefit of more information outweighs the increased context length in our framework.
>
> ### **Requested Changes 3. Effects of maximum LLM context length**
>
> > Also, the term "Adaptive Token Reduction" is misleading here. This is different from token reduction methods such as token merging, token pruning, and token learning. This is rather "context reduction" in text space rather than in token space. I would ask the authors to reconsider the naming here for better clarity.
>
> We appreciate the reviewer’s insightful comment and agree that the term “Adaptive Token Reduction” may cause confusion. To better reflect its actual functionality and improve clarity, we will rename it to **"Adaptive Context Reduction"** throughout the revised manuscript.

---

> ### Author Response · Authors · 2025-11-17
> **Response to Reviewer d1wR (Part 3/5)**
>
> ### **Requested Changes 4. More comparisons to baselines**
>
> > I see that the proprietary models considered here are somewhat outdated. I understand the field is moving really fast, and it's hard to always keep up. But, it would be nice to see at least some minimal comparisons with most up-to-date models (e.g., Gemini-2.5, Qwen3-VL).
>
> We compare our method with Gemini 2.5-Flash, Qwen2.5-VL 7B, Qwen2.5-VL 72B,  Qwen3-VL 8B, Qwen3-VL 32B and Video-R1 on Video-MMMU, Video-MMLU, Video-MME and CGBench-Reasoning. The results are shown in the table below.
>
> |                  |                            |            |                   |                          |
> | ---------------- | -------------------------- | ---------- | ----------------- | ------------------------ |
> | Model            | Video Reasoning Benchmarks |            |                   | General Video Benchmarks |
> |                  | Video-MMMU                 | Video-MMLU | CGBench-Reasoning | Video-MME                |
> | _Proprietary_    |                            |            |                   |                          |
> | Gemini 2.5-Flash | 79.2                       | -          | -                 | 81.5                     |
> | Gemini 2.5-Pro   | 83.6                       | -          | -                 | 86.9                     |
> | _Open-source_    |                            |            |                   |                          |
> | Qwen2.5-VL 7B    | 50.4                       | 32.9       | 32.6              | 71.6                     |
> | Qwen2.5-VL 72B   | 61.0                       | 40.5       | -                 | **79.1**                 |
> | Qwen3-VL 8B      | 72.8                       | -          | -                 | 71.8                     |
> | Qwen3-VL 32B     | 79.0                       | -          | -                 | 77.3                     |
> | Video-R1         | 52.4                       | -          | 30.1              | 61.4                     |
> | SiLVR (ours)     | **82.7**                   | **83.1**   | **53.4**          | 74.2                     |
>
> These results show that SiLVR is highly effective. On the critical reasoning benchmarks (Video-MMMU, Video-MMLU, CGBench-Reasoning), **SiLVR significantly outperforms every other open-source model**, including the 72B-parameter Qwen2.5-VL, the new Qwen3-VL 32B model and also the recent Video-R1. Furthermore, SiLVR's performance on Video-MMMU (82.7%) is highly competitive with the SOTA proprietary model, Gemini-2.5-Pro (83.6%), demonstrating its capability on complex reasoning tasks. These results further validate our framework’s effectiveness.
>
> ### **Requested Changes 5. Comparison with DeepSeek V3**
>
> > In the experiment in Table 2, I wonder if the comparison between Llama4 and Deepseek-R1 is the best way to show the impact of reasoning capability, because there are many differences between these two models. To make a better comparison and isolate the reasoning capability here, does it make sense to use a comparable non-reasoning model (not sure with Deepseek-V3 should be the comparison here-- Table 9 shows it is fairly-matched with R1, or using a model that supports turning off reasoning tokens)? Otherwise, the claim here is not well-supported.
>
> We thank the reviewer for this thoughtful suggestion. To better isolate the effect of reasoning capability, we have compared DeepSeek R1 with its non-reasoning counterpart, DeepSeek V3, in the table below.
>
> |              |                            |            |          |          |                          |          |          |          |
> | ------------ | -------------------------- | ---------- | -------- | -------- | ------------------------ | -------- | -------- | -------- |
> | Dataset      | Video Reasoning Benchmarks |            |          |          | General Video Benchmarks |          |          |          |
> |              | Video-MMMU                 | Video-MMLU | MMVU     | MMWorld  | VideoMME                 | CGBench  | EgoLife  | CinePile |
> | DeepSeek-V3  | 65.7                       | 74.7       | 62.9     | 58.7     | 75.0                     | 50.2     | **42.2** | 56.7     |
> | DeepSeek-R1  | **82.7**                   | **83.1**   | **68.2** | **59.9** | **77.7**                 | **59.4** | 42.0     | **59.4** |
> | Average Gain | +8.0                       |            |          |          | +3.75                    |          |          |          |
>
> From the table above, we can see that using DeepSeek-R1 leads to much larger performance gains on the reasoning benchmarks, where DeepSeek-R1 surpasses DeepSeek-V3 by 8.0% on average on all video reasoning benchmarks. In contrast, on general video benchmarks, the improvements over DeepSeek-V3 are much smaller (i.e., average improvement of 3.75%). This observation is consistent with the results reported in Table 2, further highlighting the enhanced reasoning capability in our framework.

---

> > ### Comment · Reviewer_d1wR · 2025-11-28
> > **Follow-up**
> >
> > Please add all the new experimental results to Table 1 of revised manuscript, I see that some numbers are missing there.
> >
> > Also, please include a discussion on why the proposed method is behind on General video reasoning (Video-MME) compared to other SOTA.

---

> ### Author Response · Authors · 2025-11-17
> **Response to Reviewer d1wR (Part 4/5)**
>
> ### **Requested Changes 6. More discussion about reasoning entirely in language space**
>
> > Although the proposed multi-stage pipeline provides more-flexibility to use dedicated models (captioner, ASR, reasoner), it may also suffer from loss of information (particularly, in vision -> text) compared to a multi-modal single-stage model. I would like more discussion on this to be included in the paper. The authors also highlight the benefit of “Performing reasoning entirely in language space”. Is this inherently better than natively multi-modal reasoning?
>
> We thank the reviewer for raising this fundamental point, as it touches on the **core trade-off** between language-centric pipelines and end-to-end multimodal models.
>
> **The information loss trade-off.**
>
> We acknowledge that converting vision to text will inevitably trade some fine-grained, pixel-level _spatial_ detail. However, we argue this might be a beneficial trade-off to the one made by end-to-end models. Specifically, due to prohibitive GPU memory constraints, the end-to-end video models must use extremely sparse frame sampling (e.g., 64 frames for an hour-long video). In contrast, our framework encodes video into a compact text representation, which enables **much denser temporal sampling**. Our strong empirical performance across multiple benchmarks suggests that for complex _reasoning_ tasks, this **rich temporal fidelity is far more critical** than the pixel-level spatial detail we trade away. Furthermore, our design is extensible: fine-grained visual cues (like bounding boxes or object-level descriptions) can be incorporated as additional text in the future versions of our framework.
>
> **Performing reasoning entirely in language space.**
>
> Compared to natively multimodal reasoning models, reasoning entirely in the language space offers several advantages.
>
> Firstly, it enables direct integration with very powerful LLM reasoning models (e.g., DeepSeek-R1) for complex video understanding. In contrast, multi-modal single-stage models cannot easily support large reasoning LLMs (e.g., DeepSeek-R1) due to GPU memory constraints.
>
> Secondly, this design makes SiLVR exceptionally modular and flexible. It is a training-free framework that can **"ride the wave" of progress** in foundation models. As new, more powerful video captioners or LLMs are released, they can be integrated seamlessly to instantly improve performance. This modularity also allows for the straightforward incorporation of new modalities and task-specific information (e.g., speech transcripts, object detections) without the complex, resource-intensive training required by other paradigms.
>
> Finally, we wish to position SiLVR as a **powerful alternative and challenging baseline** for current video-native reasoning paradigms (e.g., Video-R1). While we agree that end-to-end multi-modal single-stage models are a promising direction and might eventually prevail, our work demonstrates that a modular, language-centric framework offers a highly competitive, flexible path to complex video understanding. SiLVR's strong performance underscores the viability of this approach, and we view it as a critical baseline that future multi-modal single-stage systems should be measured against. We believe its simple yet effective design offers valuable insights that can help inspire new directions in both language-driven and video-native video understanding research.
>
> ### **Requested Changes 7. Details about how ASR is segmented.**
>
> > How is ASR segmented (sentence-level, ...)? Please provide more details.We apologize for the confusion.
>
> We directly use Whisper's raw output for ASR segmentation. In practice, one sentence may be divided into multiple shorter phrases, each with its own timestamp. We keep the timestamps before each phrase because it improves overall performance.
>
> For example, a typical ASR output is formatted as follows:
>
> 00:00:00,080 --> 00:00:05,120 in this video I'm going to show you
> \
> 00:00:01,640 --> 00:00:05,120 a better way to clean your keyboard

---

> ### Author Response · Authors · 2025-11-17
> **Response to Reviewer d1wR (Part 5/5)**
>
> ### **Requested Changes 8. LLM sizes.**
>
> > Please report LLM model sizes in all tables, as it gives more context when comparing different methods.
>
> In the revised version, we will report LLM model sizes in all tables to make the comparisons more transparent.
>
> ### **Requested Changes 9. Clarification on Table 6.**
>
> > Table 6: Speech is definitely more semantic/compressed than visual captions, and contains high-density ground-truth information compared to generated noisy information. This may be discussed with the results in this table. Also, please report the context length used in each setting in this table to highlight that it is a fair comparison.
>
> We have added the context length for each setting in Table 6 to ensure a fair comparison. The updated table is shown below:
>
> | Dropping Rate  |          |                        | Accuracy |
> | -------------- | -------- | ---------------------- | -------- |
> | Subtitles      | Captions | Average Context Length |          |
> | 50%            | -        | 4.3k                   | 65.3     |
> | 75%            | -        | 2.6k                   | 56.0     |
> | -              | 50%      | 7.2k                   | 68.9     |
> | -              | 75%      | 6.0k                   | 67.7     |
> | No Compression |          | 9.3k                   | 70.3     |

---

> ### Comment · Reviewer_d1wR · 2025-11-28
> **All my concerns are addressed. I recommend acceptance.**
>
> As per title.
> I thank the authors for their extensive effort in this rebuttal.

---

> > ### Author Response · Authors · 2025-11-29
> > **Response to Reviewer d1wR**
> >
> > We thank the reviewer for raising the score and for their positive acknowledgement of our extensive efforts during the rebuttal. We greatly appreciate these final constructive suggestions for finalizing the manuscript.
> >
> > Regarding the experimental results, we confirm that all new data has been incorporated into the revised manuscript. We have updated Table 1 with the new results for Video-MMMU and Video-MMLU. Due to space constraints in the main paper, the comprehensive results for Video-MME (overall) and CGBench-Reasoning are detailed in the supplementary materials (Tables 17 and 16, respectively). Furthermore, as requested, we will include a discussion in the final draft analyzing the performance comparison on the general Video-MME benchmark.
> >
> > We thank the reviewer again for their time and valuable input throughout the review process.

---

### Author Response · Authors · 2025-11-25
**General Response**

Dear Reviewers and Action Editor,

Thank you for your valuable feedback. We have revised the manuscript to incorporate the promised updates and address the reviewers’ comments. The revised sections are highlighted in blue for your convenience.

We appreciate your time and consideration, and we look forward to your further feedback.

Best,
Authors

---

### Decision · Action_Editor_uiyJ · 2025-12-30

**Recommendation:** Accept as is

**Audience:**

Yes

**Audience Explanation:**

Yes, the subset of the TMLR audience that are interested in VLMs, long-context and video understanding will find this paper of interest. Video understanding is popular topic in the community, and although the paper is not particularly novel given that prior works exist that represent a video as text, and then reason over the text, the AE believes that the paper would still be of interest to individuals in the TMLR community. The claims made in the paper are also substantiated clearly, which means it will be useful to the community.

**Claims And Evidence:**

Yes

**Claims Explanation:**

The authors propose a simple video reasoning framework that consists of two main stages: 1) Representing the video and associated audio as text via captioning and ASR transcription and 2) performing reasoning on this textual representation of the video. Although there are numerous prior works that also represent videos as text for reasoning, the authors propose an adaptive token reduction strategy to deal with long contexts for any given video.

The authors conducted thorough experiments, ablating their method comprehensively, and applying their method to a number of video understanding benchmarks and showing meaningful improvements. The authors also comprehensively answered the reviewers' questions and concerns, and updated their paper accordingly.

Therefore, the claims in this paper are well substantiated.